

# Tropospheric ozone column dataset from OMPS-LP/OMPS-NM limb-nadir matching

Andrea Orfanoz-Cheuquelaf[1], Carlo Arosio[1], Alexei Rozanov[1], Mark Weber[1],
Annette Ladstätter-Weißenmayer[1], John P. Burrows[1], Anne M. Thompson[2,3], Ryan M. Stauffer[2], and
Debra E. Kollonige[2,4]

[1]Institute of Environmental Physics, University of Bremen, Otto-Hahn-Allee 1, D-28359 Bremen, Germany
[2]Earth Sciences Division, NASA/Goddard Space Flight Center, Greenbelt, MD, USA
[3]GESTAR and Joint Center for Earth Systems Technology, University of Maryland, Baltimore County, Baltimore, MD, USA
[4]Science Systems and Applications, Inc., Lanham, MD, USA

**Correspondence:** Andrea Orfanoz-Cheuquelaf (andrea@iup.physik.uni-bremen.de)

**Abstract.**

A Tropospheric Ozone Column (TrOC) dataset from the Ozone Mapping and Profiler Suite (OMPS) observations was generated by combining the retrieved total ozone column from OMPS - Nadir Mapper (OMPS-NM) and limb profiles from OMPS - Limb Profiler (OMPS-LP) data. All datasets were generated at the University of Bremen, and the TrOC product was obtained by applying the Limb-Nadir Matching technique (LNM). The retrieval algorithm and a comprehensive analysis of the 5 uncertainty budget are presented here. The OMPS-LNM-TrOC dataset (2012-2018) is analysed and validated by comparing with ozonesondes, tropospheric ozone residual (TOR) data from the combined Ozone Monitoring Instrument/Microwave Limb Sounder (OMI/MLS) observations, and the TROPOspheric Monitoring Instrument (TROPOMI) Convective Cloud Differential technique (CCD) dataset. The OMPS-LNM TrOC is generally lower than the other datasets. The average bias with respect to 10 ozonesondes is $-1.7$ DU with no significant latitudinal dependence identified. The mean difference with respect to OMI/MLS TOR and TROPOMI CCD is $-3.4$ and $-1.8$ DU, respectively. The seasonality and inter-annual variability are in good agreement with all comparison datasets.

## 1 Introduction

Ozone ($O_3$) in the troposphere is a harmful pollutant and a near-term climate forcer (Shindell et al., 2006; Stevenson et al., 15 2013; Schultz et al., 2015). There are two sources of tropospheric $O_3$: photochemical production and transport from the stratosphere. Sources of chemical precursors of this secondary pollutant originate from biomass burning, lightning and anthropogenic emissions. Tropospheric $O_3$ concentration depends on local production and losses, as well as long-range transport (Archibald et al., 2020). Additional contributions come from intrusions of stratospheric $O_3$ (e.g. Škerlak et al., 2014). The residence time of $O_3$ in the continental boundary layer is in the range of a few days, while in the free troposphere, the annual mean lifetime 20 of $O_3$ is around 40 days (Kourtidis, 2002).



A global assessment of the amount and evolution of tropospheric $O_3$ is potentially possible by using passive remote sensing by space-borne instruments (e.g. Gaudel et al., 2018; Heue et al., 2016; Leventidou et al., 2018). The retrieval of tropospheric $O_3$ from the measurements of satellite sensors started in the late 1980s. One of the commonly used methods is the integration over the troposphere of the $O_3$ profiles retrieved from nadir measurements in the infrared (IR) or ultraviolet (UV) spectral range. Tropospheric $O_3$ from nadir profiles is available, for example, from the Infrared Atmospheric Sounding Interferometer (IASI), Greenhouse gases Observing SATellite (GOSAT), Global Ozone Monitoring Experiment 2 (GOME-2), and TROPOspheric Monitoring Instrument (TROPOMI) (Boynard et al., 2009; Ohyama et al., 2012; Miles et al., 2015; Mettig et al., 2022). The disadvantage of this approach is that the profiles may be less sensitive to changes in boundary layer $O_3$ (Doche et al., 2014). The accuracy of this technique is limited by a relatively coarse vertical resolution (7-10 km) of the retrieved profiles (see e.g. Mettig et al., 2022, and references therein).

Another method to obtain tropospheric $O_3$ columns (TrOC) from nadir-viewing satellite measurements is the Convective Cloud Differential technique (CCD). This technique has been applied to the series of Total Ozone Mapping Spectrometers (TOMS) and Global Ozone Monitoring Experiment (GOME) instruments, the SCanning Imaging Absorption spectroMeter for Atmospheric CHartographY (SCIAMACHY) and TROPOMI (Ziemke et al., 1998; Valks et al., 2014; Leventidou et al., 2016; Ziemke et al., 2019; Hubert et al., 2021; Heue et al., 2021). In simple terms, this method subtracts the $O_3$ column retrieved above clouds from that for clear-sky scenes to obtain tropospheric $O_3$. The drawback here is the requirement of a zonal invariance of stratospheric $O_3$, which is reasonably well fulfilled only in the tropics. In addition, the ozone column is obtained up to the cloud-top level, which is generally well below the tropopause.

In 1987, Fishman and Larsen (1987) proposed a residual technique that subtracts the stratospheric amount of $O_3$ from its total column using observations from different instruments (tropospheric $O_3$ residual, TOR, technique). While the total $O_3$ column is retrieved from nadir measurements, its stratospheric contribution is obtained from limb observations providing $O_3$ profiles at a much higher vertical resolution (e.g. Ziemke et al., 1998; Fishman and Balok, 1999; Ladstätter-Weißenmayer et al., 2004; Ziemke et al., 2006; Schoeberl et al., 2007; Ziemke et al., 2011). The TOR technique generally provides global coverage. However, if limb and nadir instruments from different platforms are used, the stratospheric and total $O_3$ columns (locally separated) need to be averaged monthly or, in the best case, daily to achieve a global sampling which is similar to the CCD method.

Ebojie et al. (2014) were the first to use nadir and limb observations from the same instrument, SCIAMACHY (2002-2012), to obtain tropospheric ozone columns. The main advantage of this approach, referred to as the Limb-Nadir Matching (LNM), is that stratospheric and total ozone columns were obtained for nearly the same air mass, which was observed by SCIAMACHY in the limb and nadir viewing geometries within a few minutes. This technique minimizes the instrument-related bias and improves the spatiotemporal sampling.

Similar to SCIAMACHY, a combination of the Limb Profiler (LP) and Nadir Mapper (NM) instruments, which are parts of the Ozone Mapping and Profiler Suite (OMPS) on Suomi National Polar-orbiting Partnership platform (Suomi-NPP), provides the capability to observe the same air mass in limb and nadir geometries within a short time. Applying the LNM technique to the measurements from these instruments, tropospheric $O_3$ has been retrieved as described in this paper. Similar retrieval meth-





ods for stratospheric and total column ozone as for SCIAMACHY are applied to OMPS-LP and OMPS-NM measurements, respectively. The OMPS tropospheric ozone data set starting in 2012 can be merged with the 10-year tropospheric ozone time series from SCIAMACHY (2002-2012) to obtain a long-term data record of tropospheric ozone. This, however, will be a part of a follow-up study.

The OMPS instrument, as well as the ozone column and profile data used in the TrOC retrieval, are presented in Section 2. The approach to obtain TrOC is described in Section 3. An extensive analysis of the uncertainty of the TrOC dataset is given in Section 4. The OMPS-LNM-TrOC data is evaluated in Section 5 by analysing global patterns and comparing with ozonesondes and two independent satellite datasets.

## 2   Instrument and data used in the retrieval

OMPS is one of five instruments on board the Suomi-NPP satellite. The latter is a part of the Joint Polar Satellite System Program (JPSS), a collaborative program between the National Oceanic and Atmospheric Administration (NOAA) and the National Aeronautics and Space Administration (NASA) (Goldberg and Zhou, 2017). The satellite was launched on October 28th, 2011, into a sun-synchronous orbit with an ascending node at 13:30 local time at the equator. It flies at a mean altitude of 824 km (low-earth orbit) and performs about fourteen orbits per day.

OMPS comprises three instruments: Nadir Mapper (OMPS-NM), Nadir Profiler (OMPS-NP) and Limb Profiler (OMPS-LP). The detectors are focal plane arrays of two-dimensional charge-coupled devices with one spatial and one spectral dimension. OMPS-NM is a spectrometer designed to retrieve total column ozone. The spectrometer registers backscatter solar radiation from 300 to 380 nm, with a spectral resolution of 1 nm and a sampling of 0.42 nm. The footprint of OMPS-NM is approximately $50 \times 2800$ km$^2$, with a FOV of 0.27° ($\sim$ 50km) along-track and 110° across-track swath, divided into 36 bins (ground

pixels). The across-track FOV is 20 and 30 km for the two central pixels, about 50 km for other near-central pixels and increases towards the edges of the across-track swath (Flynn et al., 2004, 2014; Seftor et al., 2014). A detailed instrument description can be found in Flynn et al. (2014).

The OMPS-LP instrument was designed to retrieve vertical ozone profiles in the upper troposphere and the stratosphere. It makes observations with three vertical slits, the central one views along the satellite's orbital plane, and the other two sideways

with their tangent points (TP) located approximately 250 km apart across-track. The central slit is aligned to observe the same air masses as in the nadir viewing geometry about 7 minutes behind. OMPS-LP performs 180 limb observations, referred to as states, per orbit from which around 140 are at solar zenith angles (SZAs) below 80°, which is used as the maximum SZA in the framework of this study. The horizontal sampling is about 3 km across-track and 150 km along-track. The spectral coverage of OMPS-LP ranges from the UV (280 nm) to the NIR (1020 nm). The charge-coupled device performs instantaneous

measurements of the entire atmosphere. Individual detector pixels observe the atmosphere in 1 km steps vertically with a field of view of 1.5 km.

This study uses the total ozone column (TOC) retrieved from OMPS-NM and OMPS-LP vertical ozone profiles to calculate the stratospheric ozone column (SOC).





## 2.1 OMPS-NM WFFA TOC

The Weighting Function Fitting Approach (WFFA) developed by us is employed to retrieve total ozone columns from OMPS-NM measurements (Orfanoz-Cheuquelaf et al., 2021). WFFA is a modification of the Weighting Function Differential Optical Absorption Spectroscopy (WFDOAS) technique, which is employed for the ozone total column retrieval from GOME, GOME-2, and SCIAMACHY (Coldewey-Egbers et al., 2005; Weber et al., 2022).

Similar to the WFDOAS technique, the WFFA algorithm approximates the measured atmospheric optical depth by a Taylor 95 expansion around a first-guess atmospheric state. The main differences between the WFDOAS and WFFA are (Orfanoz-Cheuquelaf et al., 2021):

- A zero-degree polynomial is used instead of the cubic one (as in WFDOAS). This includes the broad-band spectral signature of ozone absorption in the fitting procedure.

- The spectral window is extended to 316-336 nm (in comparison to 325 to 335 nm in WFDOAS) to reduce the impact of 100 the differential ozone absorption structure in the fit.

- Only the odd-numbered spectral points are used in the retrieval, counting from the first spectral point of the selected fitting window. This selection reduces the temperature dependency of the retrieval.

The OMPS-WFFA TOC data was validated by comparisons with collocated ground-based Brewer and Dobson measurements and four other satellite TOC datasets: OMPS L2 (McPeters et al., 2019), OMI TOMS (McPeters et al., 2015), TROPOMI 105 OFFL (Garane et al., 2019) and TROPOMI WFDOAS (Orfanoz-Cheuquelaf et al., 2021). Comparison of daily collocated data with ground-based measurements shows a mean bias below 1% for 21 out of 38 stations. For 20 stations, the standard deviations of the mean differences are under 3%. A mean bias of +0.5% and a standard deviation of 1.3% were found. All comparisons between OMPS-WFFA TOC and other satellite products are consistent with respect to the seasonality and variability with latitude. OMPS WFFA TOC presents a zero yearly global mean bias with respect to OMPS L2 product of NASA, approximately 110 0.7% with respect to OMI TOMS, -0.8% with respect to TROPOMI OFFL and -2.4% with respect to TROPOMI WFDOAS. The standard deviations of the differences are around 1.7% for all satellite validation datasets, except for OMI TOMS, for which the standard deviation reaches 3.0%. Larger differences were found for polar regions and larger SZAs. Details on the WFFA retrieval algorithm and validation of the results can be found in Orfanoz-Cheuquelaf et al. (2021).

## 2.2 IUP dataset of stratospheric ozone profiles from OMPS-LP

The algorithm for retrieving limb ozone profiles (Arosio et al., 2018; Arosio, 2019) employs the regularized inversion technique with the first-order Tikhonov constraints (Rodgers, 2000) and is similar to the SCIAMACHY algorithm (Jia et al., 2015). In this study, OMPS ozone profiles version 3.3 are used. Depending on altitude, the ozone profile retrieval uses different spectral ranges of the OMPS-LP L1 version 2.5 data. For higher tangent heights (TH), three spectral segments in the UV range are selected, while for the lower stratosphere, the visible (VIS) spectral range is used. Table 1 lists TH segments, respective 120 spectral ranges selected for the retrieval, and THs used for the spectral normalization. A polynomial is subtracted from the





**Table 1.** Details of the IUP-OMPS ozone profile retrieval V3.3: TH segments, corresponding spectral ranges, and the order of the subtracted polynomial (dash means that no polynomial is subtracted). Tangent heights $TH_{norm}$ are used to normalise the radiance in the given spectral range.

| TH segment [km] | Spectral range [nm] | $TH_{norm}$ [km] | Polynomial degree |
|---|---|---|---|
| 48 - 60 | 290 - 302 | 63.5 | - |
| 34 - 49 | 305 - 313 | 51.5 | - |
| 28 - 39 | 321 - 330 | 48.5 | 0 |
| 12 - 31 | 508 - 585 | 43.5 | 1 |
| | 600 - 628 | | |
| | 630 - 660 | | |

spectrum as a part of the overall fitting procedure in the two lowermost TH segments. The order of the polynomials is also listed in the table. Further details about the OMPS retrieval can be found in Arosio et al. (2018) and Arosio (2019).

Because of long-term stability issues in the recent years, the OMPS-LP ozone profile time series based on level 1 V2.5 data are trusted only until 2018. Currently, only measurements from the central slit are used to retrieve ozone profiles because

of remaining calibration issues related to the measurements from the side slits. More information and technical details on OMPS-LP can be found in Kramarova et al. (2018), Arosio et al. (2018), and references therein.

A comprehensive validation of ozone profiles from the previous retrieval version (IUP-OMPS V2.6) was presented in Arosio et al. (2018). In Fig. 1, a comparison with ozone profiles from Microwave Limb Sounder (MLS) is shown for IUP-OMPS V2.6 (left panel) and V3.3 (right panel) data. Both panels show relative differences between IUP-OMPS and the MLS L2 version 4.2

data as a function of altitude and latitude. Differences within ±10% are observed above 20 km for both IUP-OMPS versions. Below 20 km, the differences can reach ±30%. The main improvement is found in the lower tropical stratosphere, where V3.3 shows a better agreement with MLS. The bias increases between 30 and 35 km from 20°N to the south and below 20 km polewards of 60°S.

## 2.3 Tropopause height

In order to derive the stratospheric column from the retrieved ozone profile, the tropopause height (TPH) needs to be determined. The stratospheric column is then calculated by integrating the ozone profile from the tropopause up to the top-of-atmosphere; here, to the uppermost retrieval altitude of 60.5 km.

The most commonly used definition of the tropopause is the thermal or lapse-rate tropopause. WMO (World Meteorological Organization) (1957) defines the thermal TPH as the lowest altitude level at which the lapse rate is less or equal to 2 K/km,

provided also the average lapse rate between this level and all higher levels within 2 km does not exceed this threshold. A drawback of this definition is that it might fail in polar regions if the stratosphere is very cold. Alternatively, the dynamic

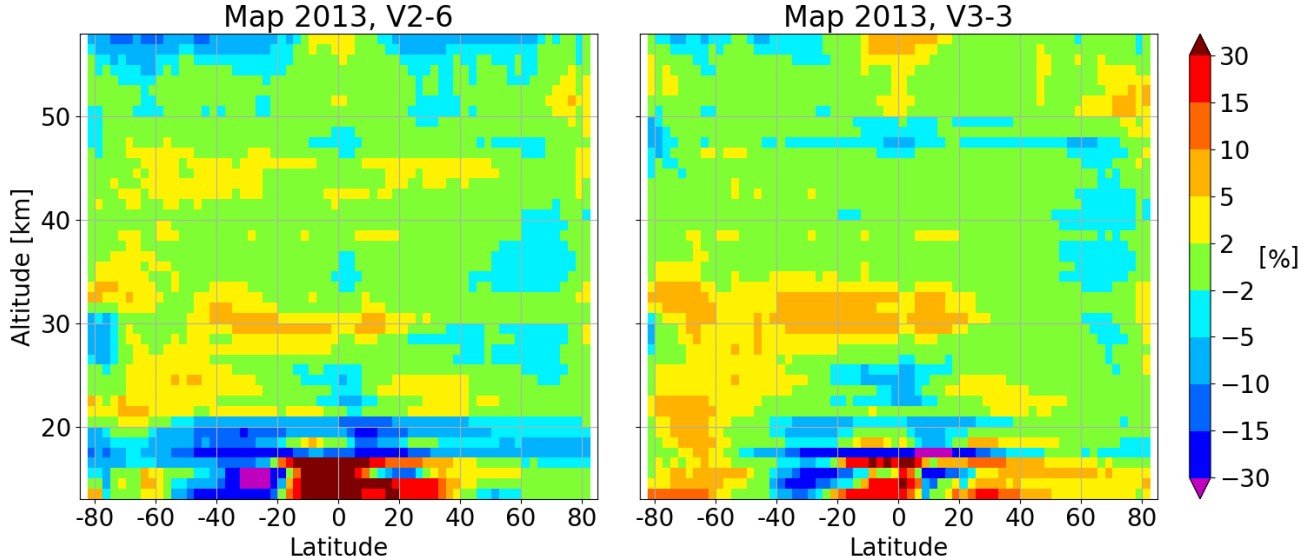

**Figure 1.** Relative mean differences between IUP-OMPS and MLS version 4.2 ozone profiles as a function of altitude and latitude. The left panel shows the comparison for IUP-OMPS V2.6 and the right panel for IUP-OMPS V3.3.

tropopause can be used, which is defined by the potential vorticity (PV), which increases with altitude. Different studies suggest this tropopause altitude to be between 1 and 4 PVU (Hoinka, 1998). This tropopause definition is, however, only applicable in the extra-tropics.

In this study, a blended tropopause definition is employed: the thermal tropopause is used in the tropics, between 20°N and 20°S, while the altitude level with a PV of 3.5 PVU (Zängl and Hoinka, 2001) is selected as TPH for latitudes above 30°. In the transition zone, between 20° and 30° latitude in each hemisphere, the TPH is calculated by averaging the thermal and the dynamical values weighted with the distance to the regime boundaries. This approach is consistent with the TPH definition employed in Ebojie et al. (2014) and Jia (2016) to obtain tropospheric ozone from SCIAMACHY using the Limb-Nadir Matching technique.

For every IUP-OMPS ozone profile, the thermal and the dynamic TPHs are determined using the ECMWF ERA-5 reanalysis data (Hersbach et al., 2020) with a spatial resolution of 0.75°×0.75° and temporal sampling of six hours.

## 3   Obtaining tropospheric ozone columns from OMPS-NM/OMPS-LP LNM

Employing the LNM technique, the tropospheric ozone product is generated for matched observations as follows:

$$\text{TrOC} = \text{TOC} - \text{SOC}. \tag{1}$$



This means that the SOC, calculated from IUP-OMPS ozone profiles, is subtracted from TOC, retrieved from OMPS-NM using the WFFA approach. The latter retrieval is done only for those across-track ground pixels that are collocated to the location of the TP of the limb ozone profiles (usually around the centre of the swath). Details on the matching procedure, which identifies the collocated ground pixels, are given below.

Although the IUP-OMPS ozone profiles are available from 8.5 km to 60.5 km altitude, only ozone values above 12.5 km are considered because of a larger uncertainty of the limb observations below this altitude. If the TPH calculated as described in section 2.3 is below 12.5 km, the ozone vertical distribution between TPH and 12.5 km is taken from the IUP-2018 ozone profile climatology (Orfanoz-Cheuquelaf et al., 2021). This climatology classifies ozone profiles as a function of season, latitude, and total ozone column. The climatological profile is offset by a scalar value which is given by the difference between

the observed and climatological ozone at 12.5 km.

    Subsequently, cloud flags provided along with the ozone profiles are analyzed. The complete profile is rejected if clouds above the TPH are detected. After passing the cloud filter, a vertical resolution quality filter is applied, which considers the mean and the standard deviation of the vertical resolutions of all profiles observed within a given calendar year. If the vertical resolution of a single ozone profile at any altitude between 12.5 and 60.5 km deviates by more than two standard deviations

from the yearly mean, the entire profile is rejected.

    The SOC in DU is determined then by integrating the $O_3$ concentration from the TPH to 60.5 km, as follows:

$$\text{SOC} = \frac{1}{2} \sum_{i(\text{TPH})}^{i(60.5\text{km})-1} \frac{(c_{i+1} + c_i)(z_{i+1} - z_i)}{2.6867 \times 10^{11}}, \tag{2}$$

where $i$ is the index of the altitude level, $c$ the ozone number density in units of $\text{molecules/cm}^3$, and $z$ the altitude in units of km.

The LNM procedure is illustrated in Fig. 2. The grid cells plotted in the figure correspond to the ground pixels of OMPS-NM, and the red points represent the tangent height footprint of the OMPS-LP states. The matching is done as follows. For each OMPS-LP state location (red points), the OMPS-NM ground pixel is identified, which contains the location of TPs (orange box in Fig. 2). Two neighbouring across-track pixels (yellow boxes) are also selected. In addition, the matching procedure also considers all triple pixels between consecutive limb states (e.g. row of three yellow boxes). The final TOC is obtained

by averaging the three OMPS-NM across-track pixels around the location of the OMPS-LP state. If the OMPS-NM pixels are located between consecutive OMPS-LP states, the final SOC value to be subtracted is obtained by interpolating between the SOCs from the two bracketing limb states. Finally, the subtraction SOC from TOC is performed to yield TrOC. The final size of the TrOC pixels is approximately 50 km along-track and 150 km across-track. For the calculation of the three-pixel mean TOC, only cloud-free OMPS-NM pixels (cloud fraction below 0.1) are used.





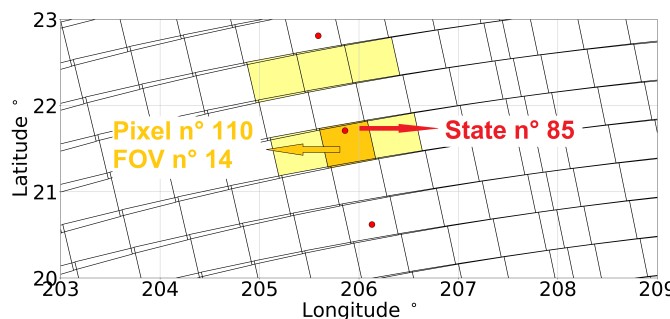

**Figure 2.** Example of a matching between OMPS-NM and OMPS-LP observation scenes. The red points mark the tangent height footprint (TPs) of the limb observations. The grid cells represent the ground pixels of OMPS-NM. The yellow and orange boxes indicate the nadir ground pixel averaged to obtain the TOC.

## 4 OMPS-LNM uncertainties

Uncertainties from the ozone total column and profile retrievals contribute to the uncertainty of the tropospheric ozone data. An additional contribution comes from the uncertainty in the tropopause height calculation. The overall TrOC uncertainty is estimated by combining these three components in a Gaussian sum as follows:

$$X_{\text{TrOC}} = \sqrt{X_{\text{TOC}}^2 + X_{\text{SOC}}^2 + X_{\text{TPH}}^2}, \tag{3}$$

where $X_{\text{TrOC}}$, $X_{\text{TOC}}$, $X_{\text{SOC}}$ and $X_{\text{TPH}}$ are the uncertainties estimated for the tropospheric ozone column, total ozone column, stratospheric ozone column, and tropopause height, respectively, all expressed in DU. All values reported in this section are assumed to be 1-$\sigma$ uncertainties and represent the uncertainty of a single observation.

The analysis in this study distinguishes between random and systematic uncertainties, with random components contributing to the variance of the data and systematic components contributing to the bias. In the following, details on the terms on the right-hand side of Eq. 3 are discussed.

### 4.1 Total ozone column uncertainty

The full analysis of uncertainties related to the WFDOAS technique is detailed in Coldewey-Egbers et al. (2005). Due to differences in the spectral window size and the polynomial degree, the uncertainty analysis was repeated for the WFFA technique for some parameters as reported in Orfanoz-Cheuquelaf et al. (2021). For some other parameters, the uncertainties derived from the WFDOAS technique were considered to be valid for WFFA as well. Table 2 summarises individual contributions to the overall TOC uncertainty.

In particular, for an increase in the tropospheric ozone by a factor of five, the uncertainty is less than $0.01\%$. For different absorption cross-sections (BDM (Malicet et al., 1995) vs Serdyuchenko (Serdyuchenko et al., 2014)), the uncertainty is less than $1\%$ for SZAs below $70°$ and increases to $2\%$ for higher SZAs. For scenes with enhanced boundary layer aerosols at



**Table 2.** Summary of contributions to the uncertainty of the total ozone column.

| Source | TOC uncertainty, % |
| --- | --- |
| Tropospheric ozone increase[1] | <0.01% |
| Ozone absorption cross-section[1] | <1% below 70° SZA |
| | 1-2% beyond 70° SZA |
| Enhanced non-absorbing aerosols[1] | 1-2% below 50° SZA |
| | ~0.5% beyond 50° SZA |
| Enhanced absorbing aerosols[1] | <1% below 50° SZA |
| | 2-3% beyond 50° SZA |
| $O_3$ and T a-priori profiles[2] | 1% below 80° SZA |
| | 5% beyond 80° SZA |
| Pseudo-spherical approximation[2] | 0.3% |

(1) Orfanoz-Cheuquelaf et al. (2021), (2) Coldewey-Egbers et al. (2005).

SZAs below 50°, the uncertainties are between 1 and 2% for non-absorbing aerosols and less than 1% for absorbing aerosols. For SZAs beyond 50°, the uncertainties are about 0.5% for non-absorbing aerosols and between 2 and 3% for absorbing aerosols. For ozone and temperature a-priori profiles, the uncertainty is 1% for SZAs under 80°. The use of the pseudo-spherical approximation results in an uncertainty of 0.3%.

From the contributions listed in Table 2, the only systematic component is the uncertainty associated with the absorption cross-section, which results in a systematic uncertainty of TOC of about 1%. The total random component is calculated by summing up all other contributions using the Gaussian rule. This results in a random uncertainty of 1.8 - 3.8%, where the range is mostly dominated by the aerosol scenarios, particularly in extreme cases. A typical random uncertainty is estimated to be about 2.8%. For a typical total ozone amount of about 300 DU (Rowland et al., 1988), the random uncertainty of 2.8% translates into 8.4 DU, and the systematic uncertainty of 1% into 3 DU.

## 4.2 Stratospheric ozone column uncertainty

A comprehensive discussion of the uncertainty budget for the IUP-OMPS ozone profiles was presented by Arosio et al. (2022). Uncertainties due to retrieval noise and from the retrieval parameters, i.e. parameters that do not enter the measurement vector, are quantified using synthetic retrievals and extensively discussed. Uncertainties originating from model approximations and spectroscopic data are also investigated. A representative set of OMPS-LP geometries was selected to provide a reliable estimation of the uncertainties as a function of latitude and season.

To assess the SOC uncertainty based on the available uncertainty budget for ozone profiles, first, we need to discuss the behaviour of the uncertainty components in the altitude domain. In this study, we consider the uncertainties of the pressure, temperature and aerosol extinction coefficient to be predominantly systematic in the altitude domain. This is because the



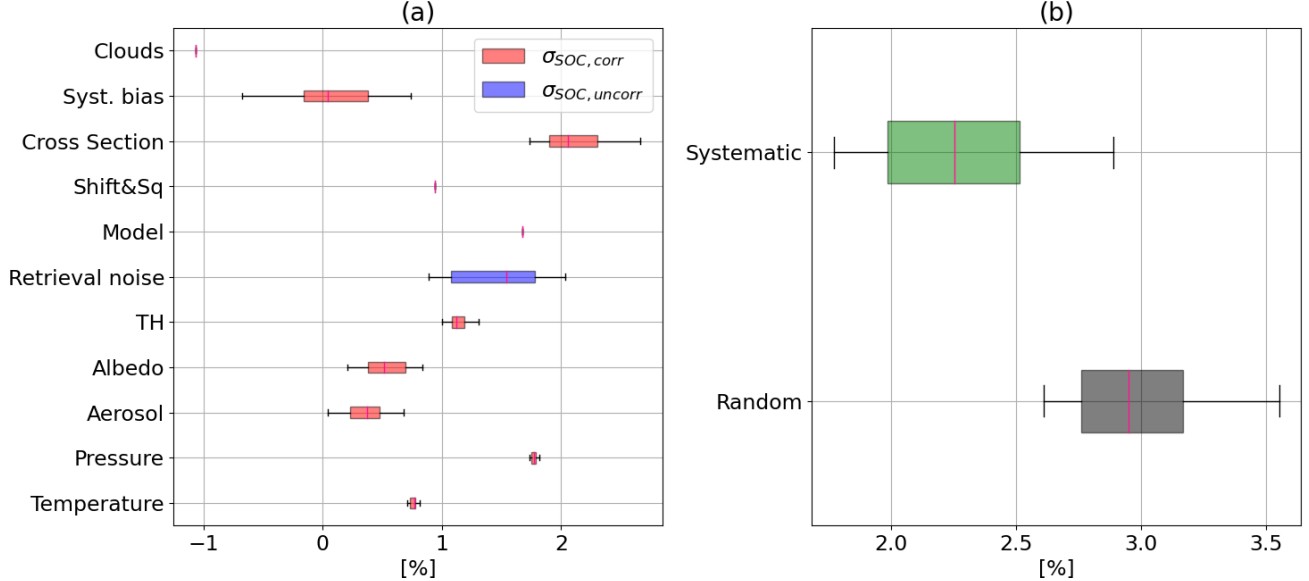

**Figure 3.** Contributions to SOC uncertainty. Panel (a): SOC uncertainties from different sources. Panel (b): total random and systematic uncertainty for SOC. The error bars span the outlier ranges, the boxes span between the first and third percentiles, and the magenta middle lines refer to the median values.

former two parameters are taken from the GEOS-5 model data, whose uncertainties originate from the model assumptions with

more probable large-scale influence rather than from random noise-like errors. Concerning the stratospheric aerosol extinction coefficients, our experience is that dominating errors in their retrievals mostly scale the resulting vertical profiles. The retrieval-noise-related uncertainty is considered uncorrelated with altitude, i.e. is randomly distributed in the vertical domain. As a consequence, the following formulas have been used to calculate the SOC uncertainty from the profile values:

$$\sigma_{SOC,correlated} = \sum_z \sigma_{O_3}(z) / \sum_z O_3(z) \tag{4}$$

$$\sigma_{SOC,uncorrelated} = \sqrt{\sum_z (\sigma_{O_3}(z))^2} / \sum_z O_3(z), \tag{5}$$

where $O_3$ and $\sigma_{O_3}$ are, respectively, the reference ozone concentrations and their uncertainties as functions of altitude $z$. The second equation was applied only for the uncertainty from retrieval noise.

The SOC uncertainty contributions estimated from a selected representative set of geometries (Arosio et al., 2022) are shown in panel (a) of Fig. 3, where different colours indicate the usage of Eq.4 (blue) or Eq.5 (red), respectively. Uncertainties related

to the cloud impact, the radiative transfer model, and the use of the spectral shift/squeeze correction in the pre-processing routine ('Shift&Sq') are independent of the viewing geometry, so that a single value is available. For other contributions, the statistics within the data sample are shown. The contribution from cloud artefacts is based on Fig. 10a of Arosio et al. (2022),





assuming that thin tropospheric clouds can still affect the retrieval bypassing the 0.1 cloud fraction threshold imposed on the nadir pixel (total column ozone) in the matching procedure.

The total random SOC uncertainty was calculated by applying the Gaussian sum:

$$\sigma_{SOC,random} = \sqrt{\sigma^2_{SOC,P} + \sigma^2_{SOC,T} + \sigma^2_{SOC,alb} + +\sigma^2_{SOC,aer} + \sigma^2_{SOC,TH} + \sigma^2_{SOC,ret.noise} + \sigma^2_{SOC,S\&S}}, \tag{6}$$

where the various terms on the right-hand side denote the individual components related to the retrieval parameters, i.e. pressure ($\sigma_{SOC,P}$), temperature ($\sigma_{SOC,T}$), surface albedo ($\sigma_{SOC,alb}$), aerosol extinction ($\sigma_{SOC,aer}$) and TH ($\sigma_{SOC,TH}$) correction, and to both the retrieval noise ($\sigma_{SOC,ret.noise}$) and the shift and squeeze correction ($\sigma_{SOC,S\&S}$).

The total systematic uncertainty was computed as follows:

$$\sigma_{SOC,systematic} = \sqrt{(\sigma_{SOC,bias} + \sigma_{SOC,clouds} + \sigma_{SOC,model})^2 + \sigma^2_{SOC,cross\ section}}, \tag{7}$$

where the terms with known signs are first summed up, i.e. the uncertainties related to the retrieval bias $\sigma_{SOC,bias}$, cloud artefacts $\sigma_{SOC,clouds}$, and radiative transfer model approximations $\sigma_{SOC,model}$. Finally, the root mean square with the cross section term $\sigma_{SOC,cross\ section}$ is calculated.

The total systematic and random SOC uncertainties are illustrated in panel (b) of Fig. 3. Their typical values are about $2.0 - 2.5\%$ and $2.8 - 3.2\%$, respectively. The median of $2.2\%$ for the systematic and $3\%$ for the random uncertainties are used below to calculate the final uncertainty of TrOC. Considering that stratospheric ozone contributes approximately $90\%$ to the total ozone, and the global average of the total ozone is 300 DU, the global average of SOC is about 270 DU. Thus, the systematic uncertainty of $2.2\%$ translates to 5.9 DU, and the random uncertainty of $3\%$ to 8 DU.

**4.3    Tropopause height uncertainty**

The calculated TPH introduces another uncertainty in the retrieved TrOC. Besides the natural variability of the TPH and the particular definition used to determine it, the uncertainty depends on the vertical resolution of the reanalysis data used to determine TPH, here ECMWF ERA-5 dataset. The ERA5 reanalysis data is provided at pressure levels, and derived quantities as the PV are defined at the centre of the layers bordering the pressure levels. To determine the tropopause altitude in km, the
pressure levels are converted into geometrical heights. In this domain, the vertical sampling of the ERA-5 dataset varies with altitude ranging from 270 to 400 m between 5 and 20 km altitude.

The TPH uncertainty as a function of latitude is determined by the vertical extent of the ERA5 layer containing the TPH and is estimated using the zonal mean (climatology) of the TPH to be 0.29 km in the northern hemisphere, 0.33 km in the tropics and 0.29 km in the southern hemisphere. To calculate the effect on TrOC, these uncertainties were added to and subtracted
from the TPH in the OMPS-LNM processing chain. The TPH range, their uncertainties, and the final contributions to the TrOC uncertainty are presented in Table 3 for three zonal bands. In the northern hemisphere (NH), the mean uncertainty in TrOC ranges from 0.8 to 2 DU, in the tropics, from 0.6 to 0.8 DU, and in the southern hemisphere (SH) between 0.8 and 1.3 DU.





**Table 3.** Uncertainties related to TPH for the northern hemisphere, tropics, and southern hemisphere.

| Latitude range | TPH [km] | TPH uncertainty [km] | TrOC uncertainty [DU] |
|---|---|---|---|
| NH ($30°$N-$60°$N) | 8.7-14.4 | 0.29 | 0.8 - 2.0 |
| Tropics ($30°$S-$30°$N) | 14.2-17.0 | 0.33 | 0.6 - 0.8 |
| SH ($30°$S-$60°$S) | 8.2-13.9 | 0.29 | 0.8 - 1.3 |

**Table 4.** Estimated uncertainties for TOC, SOC, TPH and TrOC.

| Component | TOC uncertainty [DU] | SOC uncertainty [DU] | TPH uncertainty [DU] | **TrOC uncertainty** [DU] |
|---|---|---|---|---|
| Systematic | 3.0 | 5.9 | - | **6.5** |
| Random | 8.4 | 8.0 | 0.6 - 2.0 | **12** |

### 4.4 Final tropospheric ozone column uncertainties

The systematic and random components of the final TrOC uncertainty are calculated using Eq. 3 and the results are summarised
in Table 4. The TPH contribution is considered random. Although a range of values was obtained for the TPH uncertainty, this
variation does not impact the final results. The final TrOC uncertainties do not vary significantly with latitude. The overall
random and systematic uncertainties of OMPS-LNM TrOC are about 12 DU and 6.5 DU, respectively.

The standard Level 3 product of the OMPS-LNM is a monthly mean gridded ($0.5° \times 1.5°$ (latitude/longitude)) dataset. The
overall uncertainty for a typical Level 3 data point can be estimated as follows:

$$\sigma_{TrOC_{L3}} = \sqrt{X^2_{TrOC_{syst}} + \frac{X^2_{TrOC_{random}}}{N}},$$ (8)

where $N$ is a typical number of observations averaged within a typical grid cell. In our case it is about 14. The total uncertainty
for a typical Level 3 data point is then estimated to be 7.2 DU.

### 5 OMPS-LNM tropospheric ozone evaluation

In this section, the OMPS-LNM TrOC dataset is described and compared with ozonesondes and other satellite datasets, here
OMI/MLS TOR and TROPOMI CCD. For the evaluation, the OMPS-LNM TrOC data were binned into a daily grid of $0.5° \times 1.5°$ (latitude/longitude) from $60°$S to $60°$N.

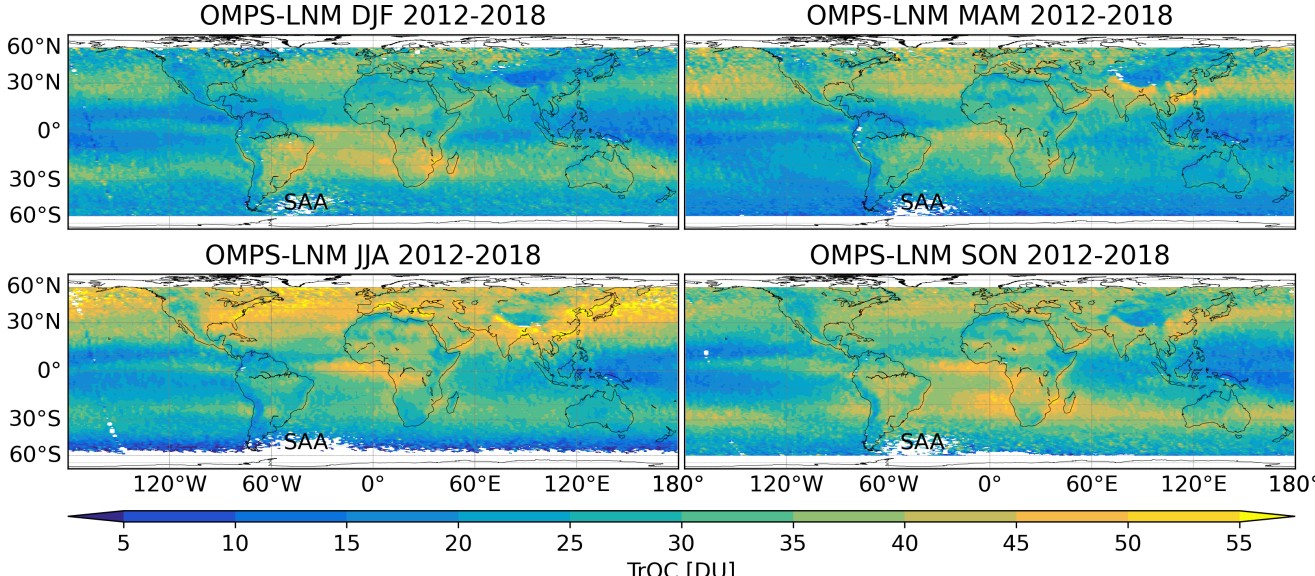

**Figure 4.** Seasonal maps of OMPS-LNM TrOC averaged from 2012 to 2018.

## 5.1 Seasonal analysis

Figure 4 shows seasonal maps of OMPS-LNM TrOC. The region of the South Atlantic Anomaly (SAA) is marked on the map. This is an anomaly of the Earth's geomagnetic field (Pavón-Carrasco and De Santis, 2016) that affects satellite electronics and

perturbs measurements. After applying the quality filters, the data density is significantly reduced within the SAA.

A band of higher values is observed between 0° to 10°N in the Pacific Ocean during all seasons and in the Atlantic Ocean during boreal summer (JJA) and autumn (SON). This feature is not seen in other satellite datasets and will be discussed in more detail in Section 5.2.

The seasonal tropospheric ozone maps show, apart from the above mentioned issue, typical features reported before. Higher

values in the extratropical hemispheric spring/summer are observed, coinciding with the likely increase in the photochemical production of tropospheric ozone (Logan, 1985). Low ozone is observed throughout the year in high-topography areas like the Andes and the Himalayas. Minimum ozone is observed above Indonesia, extending into the Pacific Ocean.

Most of the year, high values are observed in the southern subtropical Pacific Ocean (20°S-30°S), attributed to biomass burning and stratospheric intrusions (Daskalakis et al., 2022). Higher values are observed above East Asia and the northern

subtropical Pacific in all seasons, with maxima during boreal spring/summer. In addition to increased photochemical production of ozone during spring/summer, these high values might be caused by stratospheric intrusions above the ocean around 30°N (Oltmans, 2004) and outflow from intensive biomass burning in continental Southeast Asia contributing during winter and spring (Liu et al., 1999).





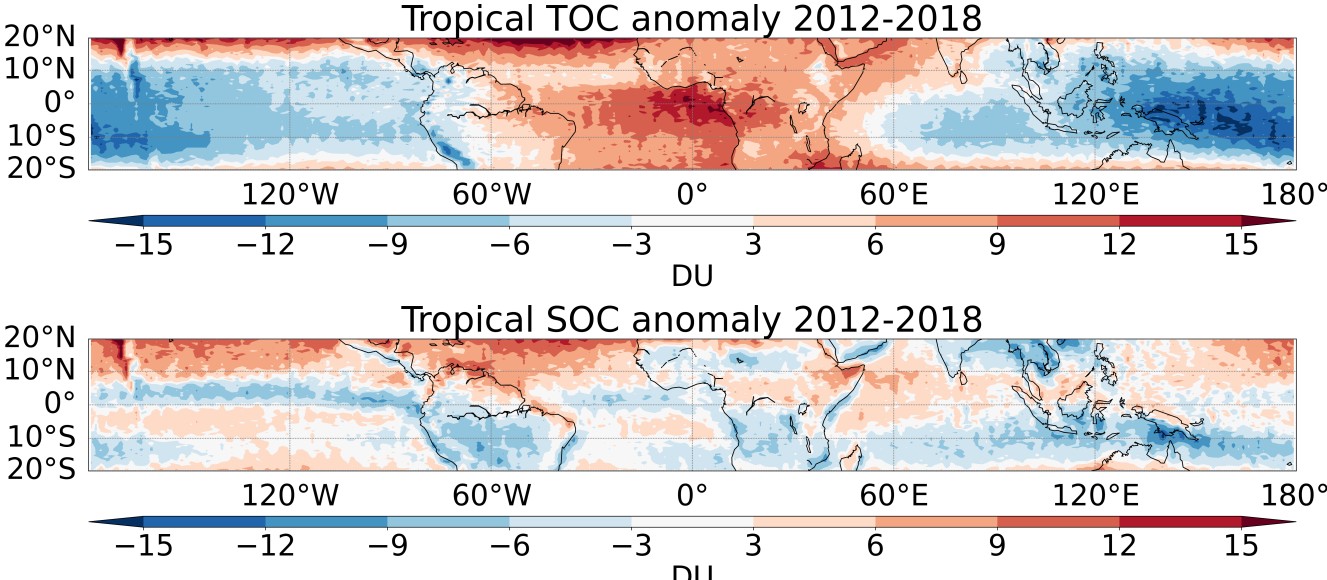

**Figure 5.** OMPS-LNM anomalies of TOC (top) and SOC (bottom) from 2012 to 2018.

High values, largest during boreal summer and lower during boreal autumn/winter, are observed over the subtropical North Atlantic throughout the year. During winter, high values are associated with the long-range transport of precursors from anthropogenic emissions in North America (Cuevas et al., 2013). In addition to the photochemical production, stratospheric intrusions are also an important contributor during the summer (Škerlak et al., 2014). Over the tropical Atlantic Ocean, lightning contributes to tropospheric ozone increase (Jenkins and Ryu, 2004). During austral spring/summer, the maximum over the southern Atlantic Ocean results from contributions of biomass burning in Africa and South America, as well as from $NO_x$ soil sources (Sauvage et al., 2007). High values above the South Indian Ocean are associated with biomass burning in Africa, particularly during austral spring, in addition to stratospheric intrusions (Fishman et al., 1991; Liu et al., 2017). The seasonality observed over the Arabian Sea is consistent with the analysis presented by Jia et al. (2017).

**5.2    Band of high ozone over the northern tropical Pacific and Atlantic oceans**

A band of fairly high tropospheric ozone columns over the Pacific and Atlantic oceans is seen in OMPS-LNM data between the Equator and 10°N. Such a band is not observed by satellite datasets which use nadir ozone profiles (Ohyama et al., 2012), the CCD method (Valks et al., 2014; Leventidou et al., 2016; Hubert et al., 2021), or residual techniques employing occultation or limb-emission measurements (Fishman et al., 2003; Ziemke et al., 2006). However, this particular feature is observable in the LNM dataset from SCIAMACHY (Jia, 2016) and in the NASA product from OMPS (Ziemke, 2019). This indicates that it is most likely a feature specific to the residual technique employing limb-scatter measurements.





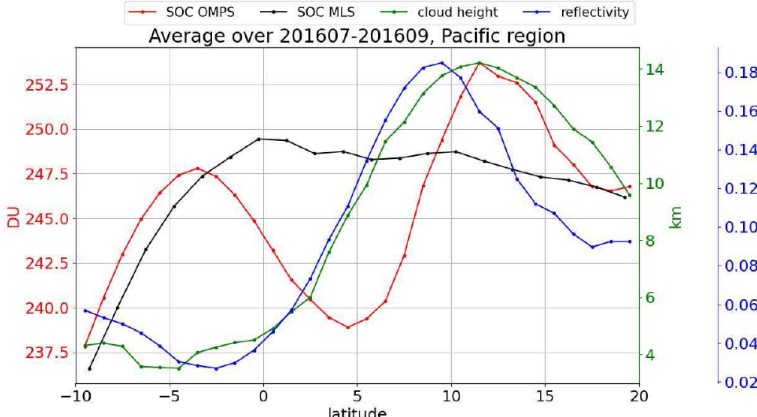

**Figure 6.** Latitudinal dependence of the July-September 2016 SOC average over the Pacific Ocean of from OMPS (red) and MLS (black), the mean cloud-top-height (green) and mean surface reflectivity (blue).

Figure 5 shows maps of TOC and SOC anomalies from the OMPS-LNM TrOC analysis between 20°S and 20°N. The anomalies were computed by subtracting the mean value from all data in the tropics. The SOC anomalies show lower values over the Pacific and Atlantic, matching the band of high TrOC. This feature is not evident in the TOC anomalies. A seasonal analysis of the SOC anomalies (not shown here) shows that this feature is persistent throughout the year, only slightly weaker during boreal summer. The negative bias in SOC leads to a high bias in the tropospheric ozone of about 10 DU (see Fig. 4). We conclude that this feature is likely an artefact from the limb ozone profiles.

Among other parameters, the surface reflectivity field retrieved along with the ozone profiles was analysed to investigate the cause of the unusually low OMPS-LP SOC. An area with higher surface reflectivity between the Equator and 10°N over the Pacific and Atlantic oceans correlates with the anomalous band. The enhanced surface albedo band is shifted a few degrees northwards and is somewhat wider than the tropospheric ozone and SOC anomaly bands. The enhanced surface reflectivity band matches the position of the Inter-Tropical Convergence Zone (ITCZ). Even though the LNM tropospheric ozone used only ozone profiles free of clouds above the tropopause and nadir pixels with cloud fraction below 0.1, the influence of the ITCZ on TrOC is evident.

Figure 6 shows the latitudinal dependence of the July-September 2016 average SOC over the Pacific Ocean from OMPS (red) and MLS (black), the cloud-top-height (green) and the surface reflectivity (blue). It is seen that the cloud top height significantly increases north of the Equator, and the reflectivity increases as well. A correlation between OMPS SOC and the reflectivity gradient is observed. When the gradient in the reflectivity is maximum, the OMPS SOC reaches a minimum, decreasing by around 10 DU relative to MLS. OMPS SOC recovers rapidly as the gradient in the reflectivity changes. The MLS SOC does not show any dependence on the presence of clouds. A similar behaviour was observed over the Atlantic Ocean (not shown here).





The current hypothesis is that a strong gradient in the surface reflectivity along the instrument Line-of-Sight (LOS) is not properly accounted for in the limb retrieval and generates artefacts in the retrieved ozone profiles. This affects all limb-scatter retrievals: IUP-OMPS, SCIAMACHY and NASA OMPS-LP. An investigation of the influence of the horizontal gradient in the reflectivity of the underlying scene on one-dimensional limb profile retrievals is out of scope of this research. However, it is important to be aware of this influence on the OMPS-LNM tropospheric ozone.

## 340    5.3    Comparison with ozonesondes

Ozonesonde data from WOUDC (Fioletov et al., 1999) and SHADOZ V6 (Witte et al., 2017; Thompson et al., 2017) were used in this study. The procedure to create collocated OMPS-LNM dataset was optimized to obtain a sufficient number of comparisons. For each ozonesonde profile, OMPS-LNM data from the grid cell enclosing the launch site and all adjacent grid cells were averaged to create the collocated OMPS dataset. The temporal averaging included OMPS-LNM data from the day

of the ozonesonde launch, one day before, and one day after the launch. Only ozonesonde sites with collocated OMPS data of at least 55 days during the entire comparison period (2012-2018) were considered. In total, 22 sites were available, eight from SHADOZ and fourteen from WOUDC. The total number of collocated days is rather low for seven years because the daily coverage of OMPS-LNM is sparse, and ozonesondes are launched four times per month at maximum.

Figure 7 shows time series of collocated ozonesonde (red) and OMPS-LNM (black) tropospheric ozone columns for three
selected sites: Madrid in the northern hemisphere (40.5°N, 3.7°W), Hilo in the tropics (19.4°N, 155.4°W) and Broadmeadows in the southern hemisphere (37.7°S, 145°E). The seasonality and variability shown by both datasets are in good agreement. The bias between the datasets is -0.8, -3.5 and -1.9 DU, respectively, with OMPS-LNM being generally lower than the sondes. The standard deviation of the difference is around 8.5 DU for each site. No clear dependence on latitude is identified from the analysis of differences between the OMPS-LNM and ozonesonde data for all sites (not shown here).

Table 5 summarises the mean value and standard deviation of the collocated data and the mean difference (OMPS-LNM - Ozonesonde) between the datasets for all compared sites. The table's first column shows the site's name and the number of collocated days. No dependency on the number of collocated days is observed. Typically, the standard deviations for OMPS-LNM are higher than those for ozonesondes by about 2.7 DU. Larger differences in the standard deviation are found for Yarmouth, Lauder, and Macquarie Island, where the standard deviations of OMPS-LNM are higher than those of the ozonesondes by 7.3,
6.6 and 6.6 DU, respectively. Some exceptions are Hilo, Natal, and Irene, where the standard deviations of the ozonesondes are higher than those for OMPS-LNM by 0.5, 0.3, and 0.5 DU respectively. The standard deviations of the differences range from 6.4 DU for Nairobi to 15 DU for Yarmouth. The bias between the datasets ranges from -6.2 DU for Irene to 2.9 DU for Hohenpeissenberg. The mean bias between OMPS-LNM and ozonesondes is found to be -1.7±2.8 DU. Seven out of 22 sites exhibit a positive bias. Eleven sites show a bias within ±2 DU.

## 365    5.4    Comparison with OMI/MLS TOR and TROPOMI CCD datasets

The OMI/MLS TOR product is a tropospheric ozone dataset retrieved using the TOR technique with TOC from OMI and SOC from MLS profiles observed since 2004 (Ziemke et al., 2006). Both instruments are aboard the Aura satellite. Measurements





**Table 5.** Comparison of TrOC from collocated ozonesonde and OMPS-LNM measurements between 2012 and 2018. The ozonesonde site location, mean values and standard deviations of the datasets as well as the mean values and standard deviation of the differences between the two datasets are listed.

| Site name (No. days) | Lat. | Lon. | sonde mean DU | sonde std. DU | OMPS-LNM mean DU | OMPS-LNM std. DU | Mean diff. (Ozonesonde - OMPS-LNM) ±1$\sigma$ DU |
|---|---|---|---|---|---|---|---|
| Legionowo (57) | 52.4°N | 21.0°E | 37.4 | 7.4 | 35.1 | 9.2 | -2.4±10.4 |
| Uccle (143) | 50.8°N | 4.3°E | 36.3 | 7.2 | 34.3 | 9.9 | -2.0±10.5 |
| Hohenpeißenberg (132) | 47.8°N | 11.0°E | 31.4 | 6.5 | 34.3 | 10.0 | 2.9±9.3 |
| Payerne (161) | 46.8°N | 6.9°E | 34.8 | 7.3 | 36.8 | 10.4 | 2.0±11.1 |
| Yarmouth (67) | 43.9°N | 66.1°W | 36.8 | 8.0 | 35.8 | 15.4 | -1.0±15.0 |
| Sapporo (61) | 43.1°N | 141.3°E | 39.6 | 9.9 | 34.0 | 15.3 | -5.7±13.2 |
| Madrid (102) | 40.5°N | 3.7°W | 35.4 | 7.3 | 34.5 | 8.6 | -0.8±8.6 |
| Boulder (61) | 40.0°N | 105.2°W | 30.4 | 5.2 | 29.7 | 7.7 | -0.7±7.3 |
| Wallops Island (88) | 37.9°N | 75.5°W | 39.6 | 8.5 | 41.1 | 12.0 | 1.5±10.5 |
| Tateno (85) | 36.1°N | 140.1°E | 39.9 | 10.0 | 34.6 | 14.2 | -5.3±11.9 |
| Naha (99) | 26.2°N | 127.7°E | 39.1 | 9.6 | 36.8 | 11.5 | -2.3±12.3 |
| Hilo (140) | 19.4°N | 155.4°W | 33.5 | 10.5 | 30.0 | 10.1 | -3.5±8.6 |
| Alajuela (63) | 10.0°N | 84.2°W | 25.0 | 6.6 | 26.4 | 8.5 | 1.4±8.8 |
| Paramaribo (68) | 5.8°N | 55.2°W | 29.7 | 6.3 | 24.3 | 8.8 | -5.3±8.6 |
| Nairobi (98) | 1.3°S | 36.8°E | 28.4 | 5.4 | 28.6 | 6.4 | 0.2±6.4 |
| Natal (72) | 5.4°S | 35.4°W | 35.9 | 8.2 | 30.8 | 7.9 | -5.1±9.2 |
| Samoa (72) | 14.4°S | 170.6°W | 21.6 | 5.1 | 23.4 | 9.2 | 1.7±8.5 |
| La Réunion Island (117) | 21.2°S | 55.5°E | 39.2 | 8.6 | 34.4 | 8.6 | -4.8±8.2 |
| Irene (65) | 25.9°S | 28.2°E | 37.8 | 7.4 | 31.6 | 6.8 | -6.2±8.9 |
| Broadmeadows (120) | 37.7°S | 144.9°E | 28.0 | 6.3 | 26.0 | 7.9 | -1.9±8.3 |
| Lauder (110) | 45.0°S | 169.7°E | 22.48 | 3.8 | 20.0 | 10.4 | -2.4±10.4 |
| Macquarie Island (57) | 54.5°S | 158.9°E | 19.8 | 4.1 | 21.6 | 10.7 | 1.8±11.4 |



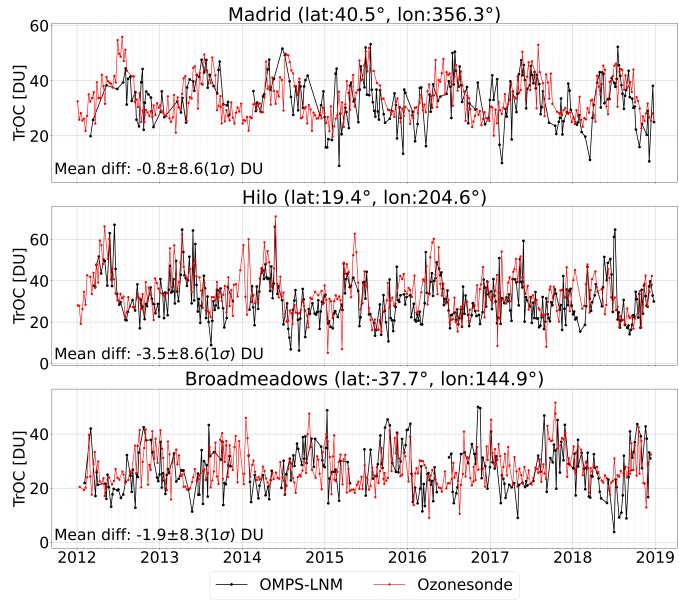

**Figure 7.** Time series of tropospheric ozone column from ozonesondes (red) and OMPS-LNM (black) for three selected sites.

of MLS are taken approximately 7 minutes before OMI. The OMI/MLS TOR product is available as monthly mean values on a $1° \times 1.25°$ (latitude/longitude) grid at https://acd-ext.gsfc.nasa.gov/Data_services/cloud_slice/new_data.html. The monthly

means are obtained by subtracting daily means of SOC from TOC daily means and subsequent averaging. The TPH needed to determine SOC was obtained using the thermal definition of the tropopause from the NCEP re-analysis. A moving 2D (latitude/longitude) Gaussian function was used to fill gaps in SOC in the along-track direction, followed by a linear interpolation along the longitude. Daily global gridded maps were generated with $1°\times1.25°$ spatial sampling. OMI TOMS L3 TOC data was filtered for clear-sky conditions by keeping only measurements with reflectivity less than 0.3. A comparison of OMI/MLS

TOR with ozonesondes from September 2004 to August 2005 showed a difference of around 2 DU, with OMI/MLS being higher than the ozonesondes (Ziemke et al., 2006). For the comparison here, the OMPS-LNM data was re-gridded onto the OMI/MLS grid ($1°$ lat $\times$ $1.25°$ lon) and averaged monthly.

The operational TrOC product from TROPOMI aboard S5P is derived with the CCD technique using the OFFL GODFIT V4 TOC (Hubert et al., 2021). The reference region, i.e. the region to estimate the above deep convective clouds column

(ACCO), is the tropical eastern Indian and western Pacific oceans, 20°S-20°N and 70°E-170°W. The deep convective clouds are selected using a cloud fraction larger than 0.8, cloud albedo higher than 0.8, and effective cloud pressure less than 300 hPa. The ozone profile climatology of McPeters and Labow (2012) is used to estimate the ozone column from the retrieved cloud top pressure to the reference level of 270 hPa. The final adjusted ACCO values are averaged over five days and 0.5° latitude bins and smoothed using a running mean over 2.5° latitude. The TOC from cloud-free pixels with cloud fraction less than 0.1

are averaged over 3-days and binned into a $0.5° \times 1°$ (latitude/longitude) grid. The ACCO is subtracted from TOC for each





grid cell. The final TrOC from the ground to the 270 hPa level is sampled daily and represents a clear-sky three-day running average (Hubert et al., 2021). Results of the validation of TROPOMI CCD by daily comparisons with SHADOZ ozonesondes from May 2018 to November 2021 are available online at http://mpc-vdaf-server.tropomi.eu/o3-tcl/o3-tcl-offl-ozone-sonde. A positive bias was found for nine SHADOZ sites, with a mean bias of 3.2 DU and a standard deviation of 1.8 DU.

In order to compare OMPS-LNM with TROPOMI CCD data, the vertical coverage of the latter dataset was extended from the 270 hPa pressure level to TPH by using the IUP-2018 ozone profile climatology. Furthermore, TROPOMI CCD data was re-grided to the finer OMPS-LNM grid and averaged monthly.

Figure 8 shows the mean difference between OMPS-LNM and TROPOMI CCD from May to December 2018 (top) and between OMPS-LNM and OMI/MLS from 2012 to 2018 (bottom). In general, the differences are negative. The overall bias is

$-1.8 \pm 4.2$ DU between OMPS-LNM and TROPOMI CCD, and $-3.4 \pm 4.7$ DU between OMPS-LNM and OMI/MLS. This is consistent with the earlier validation results for TROPOMI CCD and OMI/MLS, which were found to be on average higher than ozonesonde data.

Both comparisons show similar patterns. Positive biases are observed over South America, Middle Africa, and the Indonesian region. Over the oceans, the differences are mostly negative, between 0 and 10 DU. As discussed above, the band of positive

bias is observed between $0°$ and $10°$N over the Pacific and Atlantic oceans. The bias over the Pacific Ocean ranges from 5 to 10 DU, while over the Atlantic, it is below 5 DU. In the comparison with OMI/MLS, larger negative differences are observed in the southern extratropics. Over the northern extratropics, OMI/MLS shows higher values over Asia and North Africa, while OMPS-LNM is higher over the Atlantic Ocean. Higher values for OMPS-LNM are also observed in South America, Africa, and the Indonesian region. No seasonal variation in the differences is observed (not shown here).

Figure 9 shows monthly mean time series of TrOC (left panels) from OMPS-LNM (black), OMI/MLS (blue) and TROPOMI CCD (salmon) for six different zonal bands and differences between the OMPS-LNM and other datasets (right panels) from 2012 to 2018. Zonal bands from top to bottom are $40°$N-$60°$N, $20°$N-$40°$N, $0°$-$20°$N, $0°$-$20°$S, $20°$S-$40°$S, and $40$S$°$-$60°$S. Shadings mark the standard deviations of the averages and of the differences. In general, OMPS-LNM is lower than the other data, and the standard deviation of OMPS-LNM is higher than that of OMI/MLS. Very good agreement in the seasonal variation is observed, especially for the northern extratropics. In this latitude range, the mean difference between the OMPS-LNM and

OMI/MLS is -2.0$\pm$3.2 DU for the $40°$N-$60°$N band and -4.3$\pm$2 DU for $20°$N-$40°$N. OMPS-LNM shows a negative trend in these zonal bands, which is more clearly observed in the difference panels. In the tropics, between the Equator and $20°$N, no clear seasonality is observed for OMI/MLS. OMPS-LNM shows lower values during boreal winter, particularly low in 2015, 2016 and 2018. The TROPOMI CCD does not agree with the other two datasets in the northern tropical band. The mean bias

of OMPS-LNM in this band is -1.6$\pm$2.6 DU with respect to OMI/MLS and -2.4$\pm$2.5 DU with respect to TROPOMI CCD. From $20°$S to the Equator, OMI/MLS and OMPS-LNM agree with zero bias on average but show a difference of up to 5 DU in 2015 and 2016. In this region, the seasonal variability of TROPOMI CCD agrees with the other two datasets but shows a bias of -3.9$\pm$3.8 DU with respect to OMPS-LNM. From $20°$S to $40°$S, the seasonality of OMI/MLS data is similar to that of OMPS-LNM, and the mean bias between the datasets is -4.4$\pm$1.7 DU. In the $40$S$°$-$60°$S band, the mean difference between



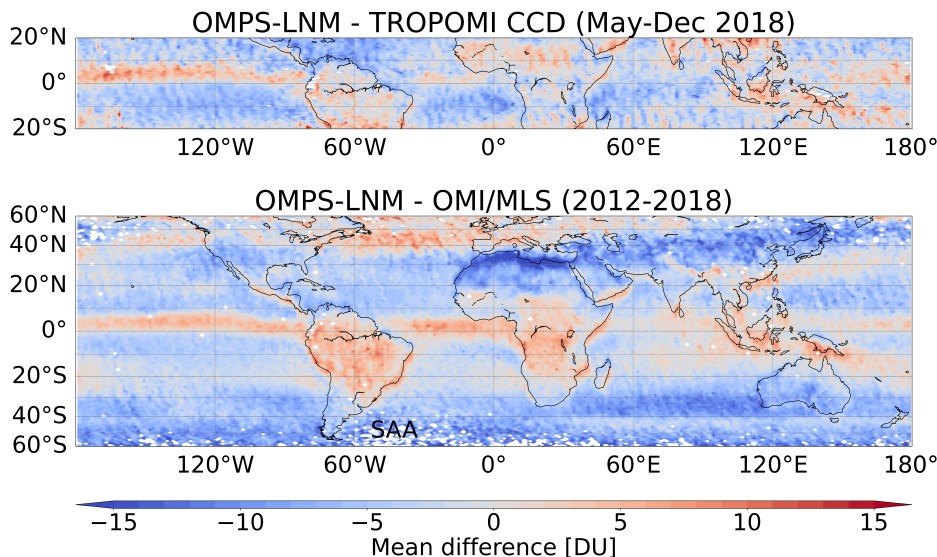

**Figure 8.** Mean differences in TrOC. Top panel: OMPS-LNM minus TROPOMI CCD from May to December 2018. Bottom panel: OMPS-LNM minus OMI/MLS from 2012 to 2018.

the datasets reaches -7.6±3.9 DU. OMI/MLS shows a much weaker seasonality in this band. In contrast, OMPS-LNM shows more pronounced minima during austral autumn and maxima during austral spring.

## 6 Summary and conclusions

This study presents a scientific tropospheric ozone column (TrOC) dataset from Suomi NPP OMPS-NM/OMPS-LP observations employing the limb-nadir-matching (LNM) technique. The data used in the retrieval, the retrieval approach, and validation results of the OMPS-LNM TrOC dataset are discussed. A detailed analysis of the uncertainties of the underlying primary data, total ozone column (TOC), tropopause height (TPH) and stratospheric ozone column (SOC), is performed, and the overall uncertainty of the final TrOC product is estimated. Systematic and random components of the uncertainty are reported. The overall systematic TrOC uncertainty is estimated to be about 6.5 DU, and the overall random uncertainty is 12 DU for a single observation.

The OMPS-LNM TrOC data was validated using ozonesonde measurements and two TrOC satellite datasets, TROPOMI CCD (Hubert et al., 2021) and OMI/MLS TOR (Ziemke et al., 2006). The comparison with measurements from 22 ozonesondes sites shows a mean bias of $-1.7 \pm 2.8$ DU with an average standard deviation of 9.9 DU. Half of the analysed sites show biases within 2 DU. We find a consistently negative bias when comparing OMPS-LNM TrOC with the two other satellite datasets. The mean bias between OMPS-LNM and OMI/MLS is $-3.4 \pm 4.7$ DU, with no seasonality in the differences. The mean bias between OMPS-LNM and TROPOMI CCD is $-1.8 \pm 4.2$ DU.



**Figure 9.** Zonal mean time series of TrOC from OMPS-LNM, OMI/MLS TOR and TROPOMI CCD (left) and differences between the OMPS-LNM and other datasets (right) for six zonal bands. The shadings indicate the standard deviations of TrOC and of the differences.



A retrieval artefact is identified over the Pacific and Atlantic oceans, showing a band of TrOC values increased by about 10 DU between the Equator and 10°N. The source for this anomaly is believed to be the impact of the gradient in the reflectivity along the instrument Line-of-Sight (LOS) on the retrieval of ozone profiles from limb-scatter measurements (OMPS-LP). The reflectivity gradient is related to a persistent belt of high clouds in the ITCZ region.

The OMPS-LNM TrOC dataset is considered to be suitable for analysing the spatial and temporal variability of tropospheric ozone and evaluating atmospheric models. It is important to consider the ITCZ's effect on the retrieval results and that the global OMPS-LNM data are, on average, 1 to 4 DU lower than other datasets considered here. This bias is, however, well within the estimated systematic uncertainty of OMPS-LNM TrOC.

*Data availability.* Our tropospheric ozone column dataset is available upon request from the University of Bremen. WOUDC ozonesonde
dataset is available at http://doi.org/10.14287/10000008, and SHADOZ dataset at https://tropo.gsfc.nasa.gov/shadoz/index.html.

*Author contributions.* All authors contributed to the design of the study. AOC developed the retrieval algorithm, performed most of the computer calculations, and made the comparisons supervised by MW, AR, and ALW. JPB provided scientific conceptual input and oversight. CA wrote and performed the calculations of section 4.2. AOC led the preparation of the paper. All authors contributed to the paper's writing and editing.

*Competing interests.* Some authors are members of the editorial board of journal Atmospheric Measurement Techniques. The peer-review process was guided by an independent editor, and the authors have also no other competing interests to declare.

*Acknowledgements.* This study was partly funded by the ESA Ozone CCI+ Phase 2 project, the University and State of Bremen. Large parts of the calculations reported here were performed at the HPC facilities of the Institute of Environmental Physics (IUP), University of Bremen, funded under the DFG/FUGG grants INST 144/379-1 and INST 144/493-1. The development of the aerosol data set required to retrieve the
ozone data used in this study was funded by the German Research Foundation (DFG) via the Research Unit VolImpact (grant no. FOR2820). The development of the stratospheric ozone profiles by Carlo Arosio was supported by his ESA Living Planet Fellowship SOLVE and the PRIME program of the German Academic Exchange Service (DAAD) funded by the German Federal Ministry of Education and Research (BMBF). Part of the data processing was done at the German HLRN (High-Performance Computer Center North). The GALAHAD Fortran Library was used for some retrievals. We acknowledge ozonesonde measurement providers and their funding agencies, the work of the PIs
and the staff of the WOUDC and SHADOZ networks.



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
