# Peer review of "Tropospheric ozone column dataset from OMPS-LP/OMPS-NM limb-nadir matching"

_Atmospheric Measurement Techniques, 2023_

## Referee Comment (RC1)

**Review of Andrea Orfanoz-Cheuquelaf et al., Tropospheric ozone column dataset from OMPS-LP/OMPS-NM limb-nadir matching**

The manuscript Tropospheric ozone column dataset from OMPS-LP/OMPS-NM limb-nadir matching by Andrea Orfanoz-Cheuquelaf et al. describes the adaption of the LNM algorithm for SCIAMACHY (Ebojie et al., 2014) to the OMPS instrument. The errors are estimated and moreover the retrieved tropospheric ozone columns are compared to ozone soundings as well as comparable satellite data sets i.e. OMI-MLS and S5P CCD.

**General Remarks / Questions**

The total column is retrieved using a WFDOAS approach (or similar) here a view more details on the algorithm might be given e.g. is a stratospheric ozone profile necessary if so which one is used? As a reader we don't know all the details that might be obvious to the authors and have been presented in previous publications.

Concerning the tropospheric ozone retrieval, are the total and stratospheric columns retrieved independent from each other, or are the retrievals linked? For example one might think of using the stratospheric profile retrieved from the limb observations in the total column retrieval, or constrain the stratospheric column by the total column as upper limit.

Only cloud free pixels (cloud fraction less than 0.1) is used. Which cloud data are used here? If only the cloud fraction is used the VIIRS cloud fraction might be an option?

The authors discuss a very interesting issue in the OMPS profile retrieval over the Pacific Ocean around 10° N and attribute it to a possible cloud effect on the profile along the Line-of-Sight (p 16 l 335). I fully agree that clarifying this issue in detail might be worth a detailed study. On the other hand as a first check it might be worth to skip the profiles being affected by clouds some hundreds of km north or south (along the Line-of-Sight) of the tangent point. This might give a first indication whether your hypothesis is correct, provided you have enough data.
This effect is only observed over the oceans but not over the continents, even though or because the convection is stronger over the continents. Is this in agreement to the current hypothesis? Unfortunately there are no ozone soundings available in the most affected regions.

The tropospheric columns agree more or less with the data from OMI-MLS or S5P_CCD. All three retrieval approaches make use of the residual technique: TrOC=TOC - SOC. Can the observed difference to OMPS-LNM be attributes to the total column or the stratospheric column. For the drift relative to OMI-MLS (p 19 l 412) this might be of interest.

Data from 2012 to 2018 are presented, however no reason is given why the following 4 years (2018- 2022) are not included, yet.

**Detailed remarks**

Figure 1 is it possible to add a mean tropopause height to the mean profile deviations. Perhaps you can add an individual profile for both MLS and OMPS (V3-3) and the difference. Because, later on you are using individual measurements.

p 6 l 145 -150: For long term time series based on SCIAMACHY and OMPS it is important to use the same definition of tropopause as in Ebojie et al.. However, this is not yet the focus of the study. Is the definition of the tropopause also consistent with the datasets used in the comparison section i.e. OMI-MLS?

p 6 l 152: Here some more details might given on the calculation of the tropopause. How are the data interpolated to the OMPS profiles / total columns.

p7 l 184: "For the calculation .. (..below 0.1) are used." this is not fully clear. Assuming one of the three pixels has a higher cloud fraction, will all three be rejected or only the cloud contaminated one. How is the spatial resolution adapted in this case.

Figure 2 Unless the data are too cloudy the tropospheric columns are retrieved along the three central columns, right?

Figure 3 One of the largest uncertainties in many satellite based trace gas retrievals are caused by the uncertainties related to the cloud fraction and altitude. Therefor the small error bars on the stratospheric ozone column with respect to the cloud data is a bit surprising. Even taken into account that only cloud fractions less than 10% are investigated the range shown here seems too optimistic.

Table 4: please add the percentile range as well. For some regions (over the tropical Pacific Ocean) the tropospheric column is as low as 20 DU (figure 4) so in this case 12 DU random uncertainty means ~60%. This is comparable to other tropospheric ozone products.

Figure 4 The figure contains some orbital structures (over the Pacific Ocean), which is a bit strange for a 6 years mean. Is this caused by the sparse spatial sampling. How many data have been averaged for the specific regions?

p 14 l 302 According to Cooper et al., 2009 Lighting NOx is an important ozone precursor over the southern US in summer and there fore contributes to the outflow over the Atlantic.

Table 5: as for table 4 add percentile deviations.

p 19 l 392 OMPS-LNM has sparse spatial coverage, has this been considered in the comparison? Comparing S5P-CCD/OMI_MLS with OMPS only where and when both datasets are available.

**Refrences:**

For your colleague Eichmann, Kai-Uwe the hyphen is missing in some references (e.g. Leventidou et al., 2018)

Cooper, O. R., S. Eckhardt, J. H. Crawford, C. C. Brown, R. C. Cohen, T. H. Bertram,
P. Wooldridge, A. Perring, W. H. Brune, X. Ren, D. Brunner, and S. L. Baughcum (2009),
Summertime buildup and decay of lightning NOx and aged thunderstorm outflow above North
America, J. Geophys Res., 114, D01101, doi:10.1029/2008JD010293

---

## Author Comment (AC1)

The manuscript "Tropospheric ozone column dataset from OMPS-LP/OMPS-NM limb-nadir matching" by A. Orfanoz-Cheuquelaf et al. presents a new research tropospheric ozone product derived from Suomi NPP OMPS satellite measurements. The paper fits to the scope of the AMT journal. The paper is well structured and written. This dataset will be of interest to the atmospheric community especially if it will be combined with a similar dataset from SCIAMACHY as promised in this publication. I recommend this paper for publication after a revision. My comments are summarized before.

**General Comment:**

It is not clear why the dataset is limited to 2012-2018 time period. Considered that Suomi NPP OMPS is still an active instrument it would be desirable to extend this dataset and provided analysis at least untill the end of 2022. That would extend the overlap with TROPOMI substantially and provide community with the up-to-date information about the status of tropospheric ozone layer.

A: The reason is explicitly mentioned in the revised manuscript:

The OMPS-LP ozone profile time series based on L1 V2.5 data, which are used by both V2.6 and V3.3 retrievals, were found to exhibit a significant positive drift after 2018 (Kramarova et al., 2018). For this reason, only the data until 2018 are used to create the OMPS-LNM-TrOC dataset. (Lines 139-141)

A new version of IUP OMPS-LP profiles is being processed based on the improved L1 (V2.6) data that counts for the observed drift after 2018. Using the improved stratospheric data, the OMPS-LNM TrOC dataset will be reprocessed, extended to the present and will be subject of a later paper. (Lines 466-468)

**Specific comments:**

p. 3 line 85. You need to rephrase this statement "The charge-coupled device performs instantaneous measurements of the entire atmosphere". I assume you meant that radiances are collected simultaneously spectrally and spatially over the FOV.

A: The text is now corrected as follows:

The pixel columns of the charge-coupled device  observe the atmosphere vertically in 1 km steps with a field of view of 1.5 km for each detector pixel. The pixel rows resister the spectral distribution of the radiance at each tangent height.  (Lines 83-85)

p.4 lines 101: Why did you include only odd-numbered spectral points and how does that relate to the temperature dependence? I assume you meant the temperature dependence of O3 cross-sections, right?

A: The line is corrected as follows:

This selection reduces the influence of the temperature weighting function within the fit procedure and makes the fit more stable. (Lines 102-103)

Details on the justification of this selection and illustration of the results are provided by Orfanoz-Cheuquelaf et al. (2021).

p.4 line104: if your intention is to name the instrument first and then the algorithm type, then it should be "OMPS-NM TOMS" rather than "OMPS L2".

A: It is now defined as "NASA's product OMPS-NM L2 V2.1" (Line 109)

Section 2.2. I was a bit confused with version numbers and references. First you refer Arosio et al., 2018, then you stated that in this study you use v3.3, but didn't provide a reference or any explanation of how that v3.3 differ from Arosio et al., 2018. It is not clear what had happened in between. Are there v2.7, 3.0 etc.? Please, clearly explain which version you use and provide the correct references. If the refernence doesn't exist, then explain how this v3.3 differ from what is described in Arosio at al., 2018.

p. 5 line 124: why can't you trust "OMPS-LP ozone profile time series based on level 1 V2.5 data" after 2018? Please provide reference or explanation. And again please clearly explain if v3.3 uses v2.5 Level 1 data or not.

P5., line 127: I didn't find any mentioning of IUP-OMPS v2.6 in Arosio et al. 2018. Are you including the right reference?

P5., line 131: Why do you see the improvements in v3.3? How does v3.3 differ from v2.6?

A: The whole section was reorganized considering answers to these questions (Lines 129-143):

This study uses OMPS ozone profiles version 3.3 (IUP-OMPS V3.3). Comprehensive validation of the ozone profiles and details about the retrieval can be found in Arosio et al. (2018) and Arosio (2019) for the previous retrieval version (here named IUP-OMPS V2.6). The main differences between V2.6 and V3.3 are in the usage of  the spectral segments and normalization THs . Table 1 lists the TH ranges, respective spectral segments selected for the retrieval, TH used for the normalization, and the order of the polynomials used for V3.3. Figure 1 presents a comparison of ozone profiles from Microwave Limb Sounder (MLS) with IUP-OMPS V2.6 (left panel) and V3.3 (right panel) data. Both panels show relative differences between IUP-OMPS and the MLS L2 version 4.2 data as a function of altitude and latitude. Differences within ±10% are observed above 20 km for both IUP-OMPS versions. Below 20 km, the differences can reach ±30%.  An overall reduction of the bias is found for IUP-OMPS V3.3, with main improvements in the lower tropical stratosphere and between 35 and 50 km. However, the bias increases between 30 and 35 km from 20°N to the south and below 20 km polewards of 60°S.

The OMPS-LP ozone profile time series based on L1 V2.5 data, which are used by both V2.6 and V3.3 retrievals, were found to exhibit a significant positive drift after 2018 (Kramarova et al., 2018). For this reason, only the data until 2018 are used to create the OMPS-LNM-TrOC dataset. Currently, only measurements from the central slit are used to retrieve ozone profiles because of remaining calibration issues related to the measurements from the side slits. More information and technical details on OMPS-LP can be found in Kramarova et al. (2018), Arosio et al. (2018), and references therein.

Figure 1. what is the time period for the comparisons? 2012-2018? Please, specify.

A: This figure shows the average for 2013. It is now explicitly stated in the figure and the caption.

P.7 lines 162-163 I assume you meant that retrieved ozone is very noisy below 12.5 km.

A: It is clarified in the revised version of the manuscript that the retrieval uncertainty is larger:

"only ozone values above 12.5 km are considered because of a large retrieval uncertainty of the limb profiles below this altitude (Arosio et al., 2018)"  (Lines 173-174)

Have you tried to use retrieved values instead of climatological? Does that change calculated SOC values in any significant way?

A: As suggested by the reviewer, we processed a sub-sample of the data set by using the retrieved values when the tropopause high is lower than 12 km (and higher than 8.5 km), instead of the climatology. Differences of around 5 DU in the SOC field are found at high latitudes when using the retrieved values below 12 km instead of the climatology (see figure below). Below 12 km, the vertical resolution of the profiles is, on average, larger than 10 km, which makes the retrieved values unreliable. The use of the climatology gives larger SOC values, leading to lower tropospheric ozone columns.

[Figure]

P.7 Lines 167. Are you deriving the cloud height from OMPS-LP? Then you should have mentioned how it's done in Sec. 2.2.

A: It is now mentioned in Lines 125-128:

Clouds in the instrument field of view are detected using the Color Index Ratio (CIR) concept described in Eichmann et al. (2016). The ratio between two radiances at wavelengths with weak ozone absorption (754 and 868 nm), called the Color Index (CI), is calculated for every TH. The CIR is defined as the ratio of the CI at two neighbouring THs. For CIR higher than 1.08, the tangent height is marked as cloudy.

P.7 line 184 Since the cloud fraction is one of the key factors in producing TrOC, it would be good if you describe how the cloud fraction is calculated in Sec. 2.1 when you talk about TOC.

R: The cloud information is now detailed: "The cloud fraction information is obtained from the operational OMPS-NM L2 product V2.1 from NASA (Jaross, 2017). The retrieval of effective cloud fraction is made using the Mixed Lambert Equivalent Reflectivity model, using a weak ozone absorption wavelength, 331.2 nm for most conditions, and 360 nm for large SZAs and high amounts of ozone (Seftor and Johnson, 2017)." (Lines 104-107).

Figure 2. All labels should be explained. What does "Pixel n$^o$ 110 and FOV n$^o$ 14" or "State n$^o$ 85" mean? Also, it might be better to say: "The red points mark the footprints of tangent points (TPs) of the limb observations".

A: The caption was modified as follows:

Example of the matching between OMPS-NM and OMPS-LP observation scenes. The red points mark the footprints of tangent points (TPs) of the limb observations. The grid cells represent the ground pixels of OMPS-NM. The yellow boxes indicate the nadir ground pixel averaged to obtain the TOC assuming the nadir pixels are cloud-free. The orange box marks the exact match between the OMPS-NM ground pixel number 110 (along-track) in position 14 (across-track) with the OMPS-LP observation number 85 (state).

Table 2. I am not sure I understand the source of error titled "Tropospheric ozone increase". First I thought it represents the error in apriori, but then I found you have another entry for "O3 and T apriori". Please, explain how the changes in tropospheric ozone affect the retrieved TOC.

A: It is clear now that it is indeed an error in the a-priori, but only in the tropospheric ozone profile.

Table 2. What is the threshold for "enhanced aerosol"?

A: Details can be found in Orfanoz-Cheuquelaf et al., 2021, as it is now mentioned in the paper (Line 2013).

The following text is from Orfanoz-Cheuquelaf et al. 2021:

We generated synthetic radiances for different aerosol scenarios using SCIATRAN V4.2 with the aerosol parameterization from LOWTRAN (Kneizys et al., 1988; Shettle and Fenn, 1979). From these radiances, the LER albedo was re trieved and used in the WFFA retrieval. The synthetic radiances were calculated with a total ozone of 325 DU, solar zenith angles of 59.88 and 27.02° (chosen from real values of OMPS-NM ground pixels), visibility of 2 km, and surface albedos of 0.05 and 0.2. The different types of boundary layer aerosols are maritime, rural, tropospheric, and urban. One case with extreme volcanic stratospheric aerosol loading was included.

p.9 line 218. I assume you mean to include a reference after "…are quantified using synthetic retrievals and extensively discussed".

A: is included " in the above mentioned study". (Line 232)

Figure 3. What did you mean by "syst. bias"? Biases in measured radiances or biases in retrieved ozone?

A: It is corrected now to "Retr. Bias", referring to the retrieval bias and also corrected in the equation.

Section 4.2. and Fig. 3. In my view several types of errors (T, P, albedo and TH) are highly correlated and it's hard to isolate contributions from them. Limb scattered radiances are directly proportional to atmospheric air density. T and P are used to calculate the density, and therefore these errors are correlated.

A: The responses in ozone profiles to these errors are indeed correlated, the errors in P, T, and TH themselves not. We see no reasons why we cannot investigate influence of each of this error independently and then sum them up in the Gaussian sense.

I feel that TH error is perhaps the leading source of uncertainties in limb scattering measurements. I am not sure what uncertainties in TH you are assuming to get the TH error. I believe that the TH error has two components systematic and random.

A: The error in TH used for the error estimation, i.e. ±100 m, is based on the information from the NASA team and corresponds to the best current knowledge. Contrary to the random error, a systematic error in TH is easily identified by validation studies. For now, no indications for a systematic TH error were found.

Same is with Pressure. If sea/surface pressure is off in GEOS assimilation that would produce a systematic bias in P profiles.

A: We are not aware of any reports about a systematic error in pressure data from GEOS. Unknown errors cannot be included in the error budget.

In some conditions the two errors will cancel each other, in other cases they will amplify the resulting error.

A: This is true, this is why the errors are summed up in the Gaussian sense.

Wouldn't error in albedo be a systematic error? Albedo depends on absolute calibrations and TH accuracy.

A: No, the error in albedo is predominantly random as it is retrieved for each measurement independently. Uncertainties of the aerosol scattering has the largest contribution to the albedo error. There are no indications of a dominating systematic error neither in the calibration nor in TH nor in the albedo retrievals.

I am confused with how the errors are sorted by systematic and random.

A: They are sorted based on the best knowledge about the error nature and in some cases are educated guesses. For more details on the classification of the errors, please refer to Arosio et al. 2022.

Section 4.3. Again you are ignoring the systematic part of the error in TPH which might be much larger than the error in vertical resolution you quoted here.

A: We found no indication of a systematic error when calculating TPH. There is no way to account for unknown errors in the error budget calculation. However, there is certainly a systematic part related to the choice of the TPH definition, which shall be taken into account when comparing different data sets (see efforts in TOAR II), rather then in the uncertainty budget itself.

P.12, lines 279-281. Since you are considering only nadir pixels and illuminating cloudy pixels, you end up with the stripes of data along the orbital tracks. The statement that "you are binning data" in my view misrepresents the reality. Perhaps, it would be better to say that you map your sparce measurement on to the regular grid.

A: The text is modified as: "the OMPS-LNM TrOC data were mapped onto a regular daily grid of 0.5° × 1.5° (latitude/longitude) from 60°S to 60°N." (Lines 295-296)

Figure 5. Please, specify how these anomalies were calculated. Did you subtract the long-term annual averages or seasonal averages? Also please, add a) and b) signs to each panel.

A: "The anomalies were computed by subtracting the long-term mean from all data in the tropics." (Line 332). The figure was updated, including adding labels a and b.

Figure 6. Be more specific. I assume SOC is from OMPS-LP as well as could top height, right? Is the surface reflectivity from OMPS LP or OMPS NM?

A: The figure is now more specific, as well as the text.

p.15 lines 316-318: In the text you need to add references to the figures you meant "The SOC anomalies (Fig.5b) show lower values over the Pacific and Atlantic, matching the band of high TrOC (Fig 4). This feature is not evident in the TOC anomalies (Fig. 5a)"

A: Done (Lines 332-334)

P.15 lines 322-324. How did you calculate the surface reflectivity? Is it from OMPS LP or OMPS NM?

A: Now mentioned in the text "OMPS-NM" (Line 345).

P.16, lines 340-346. Are these weighted averages? How many adjacent pixels were used/considered? What is the range of distances between the station and co-located OMPS? How was the temporal averaging applied?

A: It is mentioned that "OMPS-LNM data from the grid cell enclosing the launch site and all immediately adjacent grid cells were averaged (without any weighting) to create the collocated OMPS dataset. The temporal averaging included OMPS-LNM data from the day of the ozonesonde launch, one day before, and one day after the launch." (Lines 362-364)

P.16, line 360. How was the bias calculated? Did you calculate the mean for each instrument and then estimated the bias? Please, explain.

A: "mean difference (average of OMPS-LNM minus Ozonesonde time series)" (Lines 376)

Figure 7. Are you showing individual measurements or weekly/monthly averages. Please, specify that in the figure caption.

A: New caption: "Time series of tropospheric ozone column from individual ozonesonde measurements (red) and daily averaged collocated OMPS-LNM (black) data for three selected sites."

P.20, line 420. From figure 7 at Broadmeadows it seems that OMPS LNM overestimates the seasonal cycle compared to sonde as well.

A: We compared the two seasonal cycles in more detail and did not find a significant difference between the two. The following figure shows the time series and the absolute difference between them.

[Figure]

P.20, conclusions, lines 433-435: I disagree with the conclusion that "no seasonality in the differences". There is a clear seasonal bias in SH between LNM and OMI/MLS shown in Fig. 9.

A: The text has been modified as "with a seasonal differences of up to 10 DU in the extratropics. Nevertheless, a good agreement in the long-term variability is observed." (Lines 456-457)

**Minor comments:**

3 line 70. Should it be "OMPS comprises of three instruments…"

A: To our best knowledge, "OMPS comprises three instruments" is the correct expression. (Line 69)

P6., line 143. You defined PV above, but never defined PVU.

A: Done (Line 153)

Figure 6. It should be "over the Pacific Ocean from OMPS"

A: Done

References:

Arosio, C., Rozanov, A., Malinina, E., Eichmann, K.-U., Von Clarmann, T., and Burrows, J. P.: Retrieval of ozone profiles from OMPS limb scattering observations, Atmospheric Measurement Techniques, 11, 2135–2149, https://doi.org/10.5194/amt-11-2135-2018, 2018.

Arosio, C., Rozanov, A., Gorshelev, V., Laeng, A., and Burrows, J. P.: Assessment of the error budget for stratospheric ozone profiles retrieved from OMPS limb scatter measurements, Atmospheric Measurement Techniques, 15, 5949–5967, https://doi.org/10.5194/amt-15-5949-2022, 2022.

Orfanoz-Cheuquelaf, A., Rozanov, A., Weber, M., Arosio, C., Ladstätter-Weißenmayer, A., and Burrows, J. P.: Total ozone column from Ozone Mapping and Profiler Suite Nadir Mapper (OMPS-NM) measurements using the broadband weighting function fitting approach (WFFA), Atmospheric Measurement Techniques, 14, 5771–5789, https://doi.org/10.5194/amt-14-5771-2021, 2021.

---

## Author Comment (AC2)

**Review of Andrea Orfanoz-Cheuquelaf et al., Tropospheric ozone column dataset from OMPS-LP/OMPS-NM limb-nadir matching**

The manuscript Tropospheric ozone column dataset from OMPS-LP/OMPS-NM limb-nadir matching by Andrea Orfanoz-Cheuquelaf et al. describes the adaption of the LNM algorithm for SCIAMACHY (Ebojie et al., 2014) to the OMPS instrument. The errors are estimated and moreover the retrieved tropospheric ozone columns are compared to ozone soundings as well as comparable satellite data sets i.e. OMI-MLS and S5P CCD.

**General Remarks / Questions**

The total column is retrieved using a WFDOAS approach (or similar) here a view more details on the algorithm might be given e.g. is a stratospheric ozone profile necessary if so which one is used? As a reader we don't know all the details that might be obvious to the authors and have been presented in previous publications.

A: We prefer to do not blow up the manuscript with the details of the retrievals already published before as it might shift the focus of the paper, which is now on presenting the new product, expressing the issue newly identified in the Limb retrieval and a complete analysis of uncertainties.

Concerning the tropospheric ozone retrieval, are the total and stratospheric columns retrieved independent from each other, or are the retrievals linked? For example one might think of using the stratospheric profile retrieved from the limb observations in the total column retrieval, or constrain the stratospheric column by the total column as upper limit.

A: Now clarified in the revised version of the manuscript: "Both retrievals are completely independent of each other. For a consistency reason,  they use  the same ozone absorption cross-sections, Serdyuchenko et al. (2014)." (Lines 170-171)

Only cloud free pixels (cloud fraction less than 0.1) is used. Which cloud data are used here? If only the cloud fraction is used the VIIRS cloud fraction might be an option?

R: The cloud information is now detailed: "The cloud fraction information is obtained from the operational OMPS-NM  L2 product  V2.1 from NASA (Jaross, 2017). The retrieval of effective cloud fraction is made using the Mixed Lambert Equivalent Reflectivity model, using a weak ozone absorption wavelength, 331.2 nm for most conditions, and 360 nm for large SZAs and high amounts of ozone (Seftor and Johnson, 2017)." (Lines 104-107). Using the VIIRS cloud product might be an option but would introduce additional uncertainties from matching the footprints of both instruments (OMPS-VIIRS)

The authors discuss a very interesting issue in the OMPS profile retrieval over the Pacific Ocean around 10° N and attribute it to a possible cloud effect on the profile along the Line-of-Sight (p 16 l 335). I fully agree that clarifying this issue in detail might be worth a detailed study. On the other hand as a first check it might be worth to skip the profiles being affected by clouds some hundreds of kms north or south (along the Line-of-Sight) of the tangent point. This might give a first indication whether your hypothesis is correct, provided you have enough data. This effect is only observed over the oceans but not over the continents, even though or because the convection is stronger over the continents. Is this in agreement to the current hypothesis? Unfortunately there are no ozone soundings available in the most affected regions.

A: The effect is clearly observed when the instrument's line of sight crosses a persistent band of clouds which is always located at nearly the same geographical position and characterized by very high probability of cloud occurrences. Skipping all measurements affected by clouds at this particular location results in a loss of the statistical representativeness of the comparison because of a very small number of the remaining samples. The effect is certainly present both over the ocean and over land; however, it is much more difficult to detect anywhere else as the clouds appear statistically random at different locations along the line of sight, and the effect is strongly smeared away when calculating monthly mean values. However, the location of the ITCZ over the Pacific is fairly stable throughout the year, leading to a strong signal in yearly averages. In addition, the surface reflectivity gradient is strongest over the oceans, which typically have a lower surface albedo value with respect to land.

The tropospheric columns agree more or less with the data from OMI-MLS or S5P_CCD. All three retrieval approaches make use of the residual technique: TrOC=TOC - SOC. Can the observed difference to OMPS-LNM be attributes to the total column or the stratospheric column. For the drift relative to OMI-MLS (p 19 l 412) this might be of interest.

A: In the particular case of OMPS-LNM compared to OMI-MLS, differences between OMPS-NM WFFA TOC and OMI TOMS are lower than 2%, with OMPS-NM WFFA being higher than OMI TOMS. No drift was observed there. [Orfanoz-Cheuquelaf, 2023].

To address this question in more detail, we performed the comparison of SOC from MLS and OMPS-LP. The figure is now included in the paper (Fig. 7). It is observed that the patterns seen in the differences between OMPS-LNM and OMI/MLS originate from the SOC field.

Data from 2012 to 2018 are presented, however no reason is given why the following 4 years (2018- 2022) are not included, yet.

A: The reason is explicitly mentioned in the revised manuscript:

The OMPS-LP ozone profile time series based on L1 V2.5 data, which are used by both V2.6 and V3.3 retrievals, were found to exhibit a significant positive drift after 2018 (Kramarova et al., 2018). For this reason, only the data until 2018 are used to create the OMPS-LNM-TrOC dataset. (Lines 139-141)

A new version of IUP OMPS-LP profiles is being processed based on the improved L1 (V2.6) data that counts for the observed drift after 2018. Using the improved stratospheric data, the OMPS-LNM TrOC dataset will be reprocessed, extended to the present and will be subject of a later paper. (Lines 466-468)

**Detailed remarks are listed in the supplement**

**Detailed remarks**

Figure 1 is it possible to add a mean tropopause height to the mean profile deviations. Perhaps you can add an individual profile for both MLS and OMPS (V3-3) and the difference. Because, later on you are using individual measurements.

A: The mean tropopause height was added to the plot as suggested. We see no goal of showing a comparison for one single profile as the observed behavior might strongly vary from profile to profile and thus any selected example is not representative of the entire dataset.

p 6 l 145 -150: For long term time series based on SCIAMACHY and OMPS it is important to use the same definition of tropopause as in Ebojie et al.. However, this is not yet the focus of the study. Is the definition of the tropopause also consistent with the datasets used in the comparison section i.e. OMI-MLS?

A: This study uses the same tropopause definition as in Ebojie et al. (Line 159); OMI-MLS uses only the thermal definition (Line 191). Larger differences observed in the extratropics between OMPS-LNM and OMI-MLS might result from the difference in the TPH definition (text added in Lines 421-422)

p 6 l 152: Here some more details might given on the calculation of the tropopause. How are the data interpolated to the OMPS profiles / total columns.

A:  The ECMWF ERA-5 data has a spatial resolution of 0.75°x0.75° and a temporal sampling of six hours. The data is linearly interpolated using the four data points around the exact given location for the two closest times to obtain the TPH at the precise time and place of every limb state. (Lines 162-164)

p7 l 184: "For the calculation .. (..below 0.1) are used." this is not fully clear. Assuming one of the three pixels has a higher cloud fraction, will all three be rejected or only the cloud contaminated one. How is the spatial resolution adapted in this case.

A: If a single cloudy pixel is detected, this one is neglected, and the average is performed. The entire matching is rejected in case of two or more cloudy TOC pixels. (Lines 197-198)

Figure 2 Unless the data are too cloudy the tropospheric columns are retrieved along the three central columns, right?

A: Three nadir pixels with the closest to  the limb observation are selected and averaged to calculate tropospheric columns. They are not necessarily the central pixels of the nadir view.

Figure 3 One of the largest uncertainties in many satellite based trace gas retrievals are caused by the uncertainties related to the cloud fraction and altitude. Therefor the small error bars on the stratospheric ozone column with respect to the cloud data is a bit surprising. Even taken into account that only cloud fractions less than 10% are investigated the range shown here seems too optimistic.

A: The estimate is based on the impact of clouds on ozone profiles, which is presented in Arosio et al. 2022, Figure 7. In that case, we tried to assess the contribution of thin unfiltered clouds on the ozone profiles, as a function of height and optical depth of the cloud. Assuming that only very thin clouds are left unfiltered, we then computed their contribution in terms of SOC, in particular considering the case of a low cloud with OD=0.1. One needs also to consider that the profiles used for the matching are cloud free above the tropopause, the cloud fraction less than 10% is referring to the nadir TOC.

Table 4: please add the percentile range as well. For some regions (over the tropical Pacific Ocean) the tropospheric column is as low as 20 DU (figure 4) so in this case 12 DU random uncertainty means ~60%. This is comparable to other tropospheric ozone products.

A: The paper's Appendix now gives a new table with percentile values for three cases, considering a tropospheric ozone column of 20, 30 and 40 DU.

Figure 4 The figure contains some orbital structures (over the Pacific Ocean), which is a bit strange for a 6 years mean. Is this caused by the sparse spatial sampling. How many data have been averaged for the specific regions?

A: The orbital structure observed over the Pacific Ocean results from a lack of coverage in the change of the orbit date, reducing the density of available data. This comes from an erroneous flagging of OMPS-LP L2 data. The following figure shows the amount of OMPS-LP profiles present in grid boxes of 20° Longitude x 5° latitude during 2016.

[Figure]

p 14 l 302 According to Cooper et al., 2009 Lighting NOx is an important ozone precursor over the southern US in summer and there fore contributes to the outflow over the Atlantic.

A: In addition to photochemical production, stratospheric intrusions (Škerlak et al., 2014) and lighting (Cooper et al., 2009) are important contributors during the summer. (Lines 317-318)

Table 5: as for table 4 add percentile deviations.

A: The table has been modified to include this information

p 19 l 392 OMPS-LNM has sparse spatial coverage, has this been considered in the comparison? Comparing S5P-CCD/OMI_MLS with OMPS only where and when both datasets are available.

A: Datasets are compared only in the grid boxes where both are available. (Line 412)

**Refrences:**
For your colleague Eichmann, Kai-Uwe the hyphen is missing in some references (e.g. Leventidou et al., 2018)

A: It was fixed

Cooper, O. R., S. Eckhardt, J. H. Crawford, C. C. Brown, R. C. Cohen, T. H. Bertram, P. Wooldridge, A. Perring, W. H. Brune, X. Ren, D. Brunner, and S. L. Baughcum (2009), Summertime buildup and decay of lightning NOx and aged thunderstorm outflow above North

America, J. Geophys Res., 114, D01101, doi:10.1029/2008JD010293

Orfanoz-Cheuquelaf, A. (2023). *Retrieval of total and tropospheric ozone columns from OMPS-NPP*, PhD Thesis, Universität Bremen, doi:10.26092/elib/2179.

---

## Author Comment (AC3)

This review is by Owen Cooper, TOAR Scientific Coordinator of the TOAR-II Community Special Issue. I, or a member of the TOAR-II Steering Committee, will post comments on all papers submitted to the TOAR-II Community Special Issue, which is an inter-journal special issue accommodating submissions to six Copernicus journals: ACP (lead journal), AMT, GMD, ESSD, ASCMO and BG. The primary purpose of these reviews is to identify any discrepancies across the TOAR-II submissions, and to allow the author teams time to address the discrepancies. Additional comments may be included with the reviews.

General comments:

This is the first paper submitted to the TOAR-II Community Special Issue, and therefore there are no other papers to compare it to at this time. The OMPS-LNM satellite product and its analysis and interpretation are similar to the discussion and analysis of several tropospheric ozone satellite products presented in the TOAR-Climate paper (Gaudel et al., 2018) published during the first phase of TOAR.

Specific comments:

Line 14

IPCC now uses "short-lived climate forcer" rather than "near-term climate forcer" (Szopa et al., 2021)

A: It is now corrected  in the revised version of the manuscript.

Line 20

Regarding the lifetime of ozone in the free troposphere, the paper by Kourtidis et al. (2002) is not a good reference as they only mention a lifetime of 40 days in the Introduction, and they don't provide a reference. A better reference is Young et al. (2013) who give a global average lifetime of 22-23 days (this includes the boundary layer and the free troposphere).

A: The text is modified as suggested: "The global average lifetime of tropospheric O3 was estimated to be 22-23 days (Young et al., 2013). " (Line 19)

Line 39

It's worth pointing out that the nadir-limb approach gives tropospheric ozone values back to 1979, which is the earliest tropospheric ozone observations from satellites (Ziemke et al., 2019). See also Figure 22 in Gaudel et al. (2018), which shows clear ozone hotspots in summer 1979.

A: It is now included in the manuscript (Line 39)

Line 58

Here you mention merging the OMPS data with SCIAMACHY to produce long-term trends since 2002. But on line 124 you say that the OMPS-LP data can only be trusted until 2018. Does this mean the combined SCIAMACHY-OMPS trend will only span 2002-2018, and that additional data after 2018 will not be possible?

A:  A new version of IUP OMPS-LP profiles is being processed based on the improved L1 (V2.6) data that counts for the observed drift after 2018. Using the improved stratospheric data, the OMPS-LNM TrOC dataset will be reprocessed, extended to the present and will be subject of a later paper. (Lines 466-468)

Line 103

Here and throughout the manuscript, "data was" should be "data were"

A: Corrected (Line 108)

Line 114

This is the first mention of IUP and it needs to be defined

A: Now defined in Line 87.

Line 160

An ozone profile climatology (year 2018) has to be used to fill the gap between the lowest level of OMPS-LP (12.5 km) and the tropopause, if the tropopause is below 12 km. It would be helpful to indicate the frequency that the climatology has to be used. For example, it seems that it would only be necessary in the extra-tropics and during the cold months when the tropopause is often below 12.5 km.

A table or simple plot indicating the percent of the profiles requiring the use of the climatology by season and latitude band would be informative.

A: For one example year (2016), 27% of the processed profiles used climatology. The latitude dependence and the seasonal distribution for the northern hemisphere can be seen in the following figure.

[Figure]

How much uncertainty is introduced by using the climatology? This topic does not seem to be included in Section 4.

A: Indeed, the usage of the climatology was not included in the error budget. It is, however, unclear how it can be done.

Line 291

Logan (1985) is a landmark paper, but in terms of describing the current global distribution of ozone, and its origins, it is now out of date. Recent TOAR papers, or IPCC AR6, or the Monks et al. 2015 paper would be good choices as additional references.

A: Included (Line 306)

Figure 4

Thank you for using the same color table as the plots of satellite ozone products in Gaudel et al. (2018), it really helps to compare features between the different products.

Line 335

This is some excellent detective work to identify the likely origin of the tropical Pacific artefact.

References:

Szopa, S., V. Naik, B. Adhikary, P. Artaxo, T. Berntsen, W.D. Collins, S. Fuzzi, L. Gallardo, A. Kiendler-Scharr, Z. Klimont, H. Liao, N. Unger, and P. Zanis, 2021: Short-Lived Climate Forcers. In Climate Change 2021: The Physical Science Basis. Contribution of Working Group I to the Sixth Assessment Report of the Intergovernmental Panel on Climate Change [Masson-Delmotte, V., P. Zhai, A. Pirani, S.L. Connors, C. Péan, S. Berger, N. Caud, Y. Chen, L. Goldfarb, M.I. Gomis, M. Huang, K. Leitzell, E. Lonnoy, J.B.R. Matthews, T.K. Maycock, T. Waterfield, O. Yelekçi, R. Yu, and B. Zhou (eds.)]. Cambridge University Press, Cambridge, United Kingdom and New York, NY, USA, pp. 817–922, doi:10.1017/9781009157896.008.

Young, P. J., Archibald, A. T., Bowman, K. W., Lamarque, J.-F., Naik, V., Stevenson, D. S., Tilmes, S., Voulgarakis, A., Wild, O., Bergmann, D., Cameron-Smith, P., Cionni, I., Collins, W. J., Dalsøren, S. B., Doherty, R. M., Eyring, V., Faluvegi, G., Horowitz, L. W., Josse, B., Lee, Y. H., MacKenzie, I. A., Nagashima, T., Plummer, D. A., Righi, M., Rumbold, S. T., Skeie, R. B., Shindell, D. T., Strode, S. A., Sudo, K., Szopa, S., and Zeng, G.: Pre-industrial to end 21st century projections of tropospheric ozone from the Atmospheric Chemistry and Climate Model Intercomparison Project (ACCMIP), Atmos. Chem. Phys., 13, 2063–2090, https://doi.org/10.5194/acp-13-2063-2013, 2013.

---

## Author Response (AR2)

*We truly appreciate the comments and corrections provided. All the mentioned points are listed below.*

**Referee#1:**

- Recently Jerry Ziemke updated the OMI/MLS data set. The new version is corrected for an instrumental drift. If the updated version was used, would that change Figure 10, where a drift between OMI/MLS and OMPS LNM is obvious.

A: From the website https://acd-ext.gsfc.nasa.gov/Data_services/cloud_slice/ hosting the OMI-/MLS, we got the following statement: "For the OMI/MLS tropospheric column ozone products listed above, a small long-term soft calibration was implemented to provide better estimates of long-term trends. This adjustment (applied in late August 2023) was necessary to update our corrections for drift error of the OMI/MLS products."

According to Ziemke et al. (2019), the drift adjustment was -1.0 DU/decade. An additional -0.6 DU/decade drift was applied last summer (Ziemke, personal communication). The updated dataset was made available months after the submission of this manuscript and is provided only on a 5°x5° grid.
Figure 10 of the manuscript shows drifts of around 5 DU (even larger) between OMI/MLS and OMPS-LNM for the compared period. A correction of OMI/MSL data by the stated instrumental drift of -1.6 DU/decade would reduce the observed drift between the data sets but not fully eliminate it.

This information is included in the manuscript:
(L. 445-449 of the new version)
A drift is observed in the differences between OMI/MLS and OMPS-LNM, stronger for mid-latitudes and in the northern hemisphere. According to Ziemke (personal communication), the OMI/MLS dataset needs to be corrected by -1.6 DU/decade. This correction would reduce the observed drift of about 5 DU for the analysed period between the data sets but not fully eliminate it. OMPS-LNM shows no drift in the ozonesondes comparison (Orfanoz-Cheuquelaf, 2023, pp. 91–92).

**Referee #2:**

Authors addressed some of referees' comments in their revision. However, I have some concerns that prevent me from accepting this paper for publication in the existing form.

- All 3 referees asked authors why the dataset ends in 2018. Authors response was: "The OMPS-LP ozone profile time series based on L1 V2.5 data … were found to exhibit a significant positive drift after 2018 (Kramarova et al.,2018). For this reason, only the data until 2018 are used to create the OMPS-LNM-TrOC dataset."
A careful examination of Kramarova et al. 2018 reveals that the study was submitted in November 2017 and documented "observed biases and seasonal differences and evaluate the stability of the version 2.5 ozone record over 5.5 years" starting from April 2012. The quoted study obviously did not analyze data after 2017 and did not make any predictions about drifts after 2018. I found this very disturbing. It is fine to say that in this tropospheric study we choose to focus on the time period from 2012 to 2018 because we didn't process or analyze data after 2018, or found unexplained drift

after 2018 etc.. But it is unacceptable to refer readers to the study that does not analyzed data after 2017.

A: Thank you for pointing this out; we realized our formulation was misleading. Kramarova et al. (2018) discussed, for the first time, the drift affecting, as you rightly say, the first 5.5 years of the OMPS-LP time series. By comparing OMPS-LP deseasonalized anomalies with MLS data, we also investigated the drift affecting our stratospheric ozone product, including data until the end of 2021. Below, we show an example of the stratospheric ozone time series in terms of monthly differences with respect to MLS in the mid-latitudes at 20.5 km. We notice that, for this case, around 2018, the drift with respect to MLS becomes stronger (see Figure below). Consequently, we decided not to update the tropospheric data set beyond the end of 2018. We plan to reprocess the whole dataset as soon as the new version of stratospheric profiles is ready, which is expected to have a mitigated drift.

The manuscript is corrected as follows:
(L. 139-146 of the new version)
Our OMPS-LP ozone profile time series V2.6 and V3.3 use L1 V2.5 data. As discussed by Kramarova et al., 2018, the ozone time series above 20 km retrieved using this L1 data exhibit significant positive drift, especially after 2016. Our later investigations show that the data after 2018 is affected even more strongly. For this reason, we decided not to continue updating the OMPS-LNM-TrOC dataset beyond the end of 2018 using profiles based on L1 V2.5 data. Currently, only measurements from the central slit are used to retrieve ozone profiles because of remaining calibration issues related to the measurements from the side slits. More information and technical details on OMPS-LP can be found in Kramarova et al. (2018), Arosio et al. (2018), and references therein. A reprocessing of the OMPS-LNM-TrOC data is planned as soon as the new version of OMPS-LP stratospheric profiles (V4.0) based on the improved L1 data (V2.6) is available.

[Figure]

- Responding to Referee #1 question authors replied: "The orbital structure observed over the Pacific Ocean results from a lack of coverage in the change of the orbit date, reducing the density of available data. This comes from an erroneous flagging of OMPS-LP L2 data".
Could you please expand on this? Is it a problem with IUP-OMPS V3.3? Or is it a problem with OMPS NM total? Is it a technical problem? Can it be fixed in future revisions?

A: The problem came from a software bug. Some orbits were missing when the date changed along the orbit in the IUP-OMPS V3.3 and, consequently, in the OMPS-LNM process. This issue was fixed, and Figure 4 was updated in the latest submitted version.

- Authors explained that the cloud fraction estimates came from the NASA OMPS NM total ozone dataset (OMPS_NPP_NMTO3_L2). Authors provided a reference to the OMPS_NPP_NMTO3_L2 dataset (Jaross 2017). Then they started to described MLER model and referred to the corresponding Readme file. I feel it's inappropriate reference since the purpose of the Readme file is to provide description of the variables reported in the files but not the methodology or the algorithm. It would be more appropriate to cite the V8 ATBD document which provides the description of cloud fraction calculations: Pawan K. Bhartia and Charles W. Wellemeyer, OMI Algorithm Theoretical Basis Document, Volume II, Chapter 2, Aug. 2002, https://eospso.nasa.gov/atbd-category/49.

A: Done (L.106 of the new version). Thank you for pointing this out.

- In response to referee #2 question about assumed uncertainties in TH, authors replied "The error in TH used for the error estimation, i.e. ±100 m, is based on the information from the NASA team and corresponds to the best current knowledge". Is it an estimate for the absolute error in TH or for a drift in TH? Can you provide a reference for this? Publications by NASA's team provided larger estimates for TH error. For instance, Moy et al. 2017: "The application of RSAS to Limb Profiler (LP) measurements from the Ozone Mapping and Profiler Suite (OMPS) on board the Suomi NPP (SNPP) satellite indicates tangent height (TH) errors greater than 1 km with an absolute accuracy of ±200 m." Then Kramarova et al. 2018: "The combined accuracy of our two altitude registration methods is about ±200 m."

A: Thank you for this remark. The NASA team uses two methodologies to verify the pointing of OMPS-LP originally obtained from the star tracker information: the two papers by Moy et al. (2017) and Kramarova et al. (2018) report and discuss the accuracy and the specifics of these two methods. The RSAS is used to control the absolute pointing with an accuracy of ±200 m, whereas the ARRM is more suitable for assessing the relative pointing with an accuracy of ±100m. A combination of the results of these two methodologies is used to determine the static ground-to-flight and intra-orbital corrections. The latter is provided daily as a function of the observation number within the orbit. The uncertainty of this correction is determined by the ARRM approach uncertainty (100 m) and propagates as a random error to the retrieval results. The uncertainty of the static correction from the RSAS method (200 m) results in a potential bias of the ozone retrieval. However, the validation studies performed so far show no indication of a potential bias in the pointing. For this reason, only random uncertainty was considered in our error estimations. As mentioned above, this uncertainty is determined by the accuracy of the ARRM method, which is "sufficient to detect 100 m changes in pointing in the absence of GPH errors." (Moy et al., 2017).

References:

Moy, L., Bhartia, P. K., Jaross, G., Loughman, R., Kramarova, N., Chen, Z., Taha, G., Chen, G., and Xu, P.: Altitude registration of limb-scattered radiation, Atmos. Meas. Tech., 10, 167–178, https://doi.org/10.5194/amt-10-167-2017, 2017

Orfanoz-Cheuquelaf, A. P.: Retrieval of total and tropospheric ozone columns from OMPS-NPP, Ph.D. thesis, University of Bremen, https://doi.org/10.26092/elib/2179, 2023.